

# Eco-physiological characterization of early successional biological soil crusts in heavily human impacted areas – Implications for conservation and succession

Michelle Szyja[1], Burkhard Büdel[1], Claudia Colesie[2]

[1]Department of Plant Ecology and Systematics, University of Kaiserslautern, Germany
[2]Department of Forest genetics and Plant Physiology, Swedish University of Agricultural Sciences (SLU) Petrus Laestadius Väg

*Correspondence to*: Michelle Szyja (michelle.szyja@web.de)

**Abstract.** Eco-physiological characterization of photoautotrophic communities is not only necessary to identify the response of carbon fixation related to different climatic factors, but also to evaluate risks connected to changing environments. In biological soil crusts (BSCs), the description of eco-physiological features is difficult, due to the high variability in taxonomic composition and variable methodologies applied. Especially for BSCs in early successional stages, the available datasets are rare or focused on individual constituents, although these crusts may represent the only photoautotrophic component in many heavily disturbed ruderal areas, like parking lots or building areas with increasing surface area worldwide. We analyzed the response of photosynthesis and respiration to changing BSC water contents, temperature and light in two early successional BSCs. One BSC was dominated by the cyanobacterium *Nostoc commune*, the other by the green alga *Zygogonium ericetorum*. Independent of species composition, both crust types had convergent features like high light acclimatization and low or no depression in carbon uptake at water suprasaturation. This particular setup of eco-physiological features may enable these communities to cope with a high variety of climatic stresses, and may therefore be a reason for their success in heavily disturbed areas with ongoing human impact. Nevertheless, a major divergence between the two BSCs was their absolute carbon fixation rate on a chlorophyll basis, which was significantly higher for the cyanobacterial crust. This study emphasizes the importance of measuring intact BSCs under natural conditions for collecting reliable data.

## 1 Introduction

In drylands, the eco-physiological characterisation of biological soil crusts (BSCs) is a useful instrument for the evaluation and prediction of ecosystem functioning under recent climate change scenarios (Maestre et al., 2011). Biological soil crusts (BSCs) are small scale communities, composed of bryophytes, lichens, cyanobacteria, green algae, heterotrophic bacteria and microfungi, within or on top of the uppermost millimetres of soil surfaces (Belnap et al., 2016), that are often used as model systems to characterize the biodiversity–ecosystem function relationship in soils (Bowker et al., 2010). Investigations



describing eco-physiological characterizations of the poikilohydric soil crust organisms, i.e. lichens, bryophytes and terrestrial cyanobacteria are relatively abundant, whereas studies on terrestrial green algae in general and cyanobacteria in temperate habitats are not. Although in most climax BSC communities physiological capacity of cyanobacteria and green algae might be negligible due to their little biomass in relation to lichens and bryophytes (Belnap et al., 2001), in early BSCs

(Fischer et al., 2010) and BSCs in arrested succession their eco-physiological efficiency might be a key driver for ecosystem functioning. Additionally, habitats where cyanobacteria dominated BSCs in a climax community have been found, e.g. in high altitudes as Hochtor in Austria or in high latitudinal areas as in Ny-Alesund in Svalbard (Williams et al., 2016).

In temperate regions, BSCs are not a typical 'steady state' vegetation type. There they are either restricted to continuously disturbed habitats or are an initial stage of succession after heavy disturbances.  Biological soil crusts are a typical element of

at least temporarily arid ecosystems (Büdel, 2001) and in ecosystems with extremes in hot or cold periods and with long-lasting snow (Büdel et al., 2014). Anthropogenic disturbances like forestry, traffic lanes or trampling (Webb and Wilshire, 1983) create new land surfaces (Walker and Willig, 1999) where microclimatic conditions can favor BSC occurrence even within temperate habitats. This is due to the fact that BSCs in general are initial colonizers of open surfaces (Fischer et al., 2010; Veste et al., 2011) where cyanobacteria and green algae appear as pioneers (Belnap, 2006). While in the natural BSC

succession, lichens and bryophytes would broaden the initial colonization (Cameron et al., 1970; Belnap and Eldrigde, 2001; Belnap, 2006), ongoing disturbance prevents proceeding recovery, thus remaining at a low-level species richness (Webb and Wilshire, 1983). Cyanobacteria and green algae are more resistant to disturbance than bryophytes and lichens, subsequently frequent disturbance can maintain a BSC at an early successional stage (Kuske et al., 2011), a phenomenon also referred to as arrested succession. Areas where vegetation remains in an arrested succession are appearing more frequently due to a

growing need for space by humans (Housman, 2006). New habitats derived from accelerating human activities are fairly understudied (Rindi, 2007; Belnap et al., 2016).

Organisms of ruderal habitats often expose specific physiological adaptations, for example high physiological plasticity (Belnap & Lange, 2001; Grime, 2012). Both, cyanobacteria and green algae are the main constituents and only primary producers during a) early succession, b) regeneration processes of habitats under constant anthropogenic pressure, and c)

BSCs in demanding habitats, e.g. hot (Büdel, 1999; Karsten et al., 2010) and cold deserts (Büdel et al., 2008; Büdel et al., 2016). Typical taxa of green algae and cyanobacteria found in early successional BSCs of the temperate zone are, for example: *Zygogonium ericetorum* (Pluis, 1994; Büdel, 2001a), *Klebsormidium* spp. (Pluis, 1994), *Microcoleus* spp. (Ashley and Rushforth, 1984; Belnap, 1996; Belnap et al., 2001) and *Nostoc* spp. (Pandey et al., 2005).

Comparing eco-physiological literature of cyanobacteria and green algae reveals that cyanobacteria are in general a well-

studied group of organisms. In the absence of other photoautotrophic organisms in BSCs, they provide the most important ecosystem services (Makhalanyane et al., 2015 and sources therein). Terrestrial cyanobacteria are important in many ecosystems, due to their ability to fix atmospheric nitrogen and sequester carbon (Dojani et al., 2011; Büdel et al., 2016) and play important roles in global biogeochemical cycles (Raven, 2012). Most studies on the ecophysiology of single cyanobacteria species or cyanobacterial dominated BSCs were carried out in desert or polar environments (Hawes et al.,





1991; Lange et al., 1992; Housman, 2006; Novis et al., 2007), rarely in temperate regions. In contrast to cyanobacteria, reliable records of eukaryotic algal groups appear to be restricted to studies in soils of temperate and alpine regions (Karsten et al., 2010). Although green algal dominated BSCs occur rarely in temperate regions, where they do, they have high soil surface coverage (Büdel et al., 2016). Green algae serve as key organisms in BSC formation in all ecosystems, especially in

temperate, arctic and high alpine regions (Büdel et al., 2016). A lack of information on green algae in general and especially on temperate, ruderal areas seems surprising because green algae occur in virtually all terrestrial habitats and manmade surfaces (Gaylarde and Morton, 1999; Tomaselli et al., 2000; Büdel, 2011). The ecosystem services provided by green algae include stabilization of soil surfaces, improvement of soil structure, soil fertility and the influencing of hydrological processes (Bailey et al., 1973; Hu et al., 2002; Reisser, 2007). Reisser et al., (2007) described the current status of eco-

physiology of terrestrial green algae as being rather based on assumptions and deductions than on experimental data. Additionally, most green algal focused studies investigated organisms after a prolonged period of cultivation (e.g. Karsten et al., 2010a, 2010b, Karsten, 2013), which could have changed their eco-physiological responses over time, as these organisms are well known to have a high acclimatization potential to differing environmental factors (Dietz et al., 2000; Nash, 2008; Colesie et al., 2012; Belnap et al., 2016).

One application for detailed eco-physiological descriptions is modelling on a global scale. Post millennial interest in carbon gain of BSCs increased (Lange and Belnap, 2016) and their CO2 exchange rates are now considered relevant even on a global scale (Elbert et al., 2012; Porada et al., 2013, 2014). Unfortunately, modern process based models as used by Porada et al. (2013; 2014) are still based on few available datasets, that cover few different BSC types, organisms, geographical regions, and climatic situations; even excluding cyanobacteria and green algal dominated BSCs. Problematic is also the

focus on isolated organisms rather than studying the whole BSC system (see Colesie et al., 2016 or sources in Elbert et al., 2012). One organism studied out of several BSC organisms does not represent the ecological response of a complete BSC (Weber et al., 2012).

The goal of this study is to present a detailed description of the eco-physiological performance of a cyanobacterial or green algal dominated BSC regarding their photosynthetic response to different water, light and temperature conditions of

temperate habitats. This allows an overview of in situ gas exchange rates correlated to local climate and therefore a suitable database for potential global scale models. The study also provides an eco-physiological dataset for BSCs in habitats with ongoing human disturbance. Additionally, it demonstrates the value of measuring BSCs as a system instead of as single components.

Two major research questions raised are: 1) What is the gas exchange rate of an intact cyanobacterial or green algal BSC?

We expect a lower gross and net photosynthesis of both crusts compared to BSCs of later successional phases as their photosynthetic performance should be lower due to less biomass or lower development stage (Belnap and Lange, 2001; Colesie et al., 2014b). At the same time, a higher physiological plasticity is predicted for cyanobacteria and green algae which would enable both organism groups to cope with a wider range of abiotic stresses.



2) To what extent can the photosynthetic rate of the BSC be delineated from single organism measurements? We expect differences between measurements of complete BSCs (with attached soil and soil organisms), of the isolated organisms, and bare soil alone. In theory, a mathematical addition of "separated organisms" plus "bare soil" should equal the complete system reading. We expect that the position and arrangement of the sample inside the measurement system, here a cuvette,

will influence the photosynthetic values, as treatment of samples will always shape their response.

## 2 Material and methods

### 2.1 Study site and organisms

Two anthropogenically impacted sites with constant disturbance were selected. Both sites were located in south western Germany (Fig. 1a) and were dominated by either green algae or cyanobacteria. The sites were 50 km apart from each other,

which excludes any macroclimatic differences. Mean annual air temperature is 9.9 °C and mean annual precipitation 741.3 mm (Weather station of the Agrarmeteorologie Rhineland–Palatinate Morlautern).

Site 1 – Mehlinger heath. The Mehlinger heathland (MH, Fig. 1b) is a former military training ground (Ruby 1979) close to the city of Kaiserslautern (49°48' N, 7°83' E). Once the military use was abandoned, a BSC dominated by green algae developed between dense heather stands, which formed as a part of the natural succession process. With 150 ha, the site is

the largest heathland system of southern Germany. It is situated 320 to 340 m a.s.l. and soils are acidic, mostly due to their origin from red sandstone of the early Triassic (Landesamt für Geologie und Bergbau, Rhineland–Palatinate). Since 2001 it has been a natural reserve with an ongoing human management regime to preserve the heathland system. Vascular plant vegetation is dominated by *Calluna vulgaris*. In general, the cryptogamic diversity concerning cyanobacteria and green algae in this habitat is poor (Stanula, 2011), with only five and one species, respectively. The sampling site is close to a look-out

point, where trampling is unpreventable. The dominant organism in the occurring BSC is the green algae *Zygogonium ericetorum* (Fig. 1c).

Site 2 – Parking lot. Cyanobacteria dominated BSC samples were collected from a parking lot (PL) at an equestrian farm (Fig. 1d) near the city of Zweibrücken (49°19' N, 7°25' E), 369 m a.s.l. The bedrock is composed of base rich limestone that originates from early Triassic with a coarse gravel overlay. Daily use by cars and trampling prevent the development of

higher vegetation. Some bryophyte species occur: *Hypnum cupressiforme* and *Tortula muralis*. The cyanobacterium *Nostoc commune* Vaucher ex Bornet & Falhault (Fig. 1e) is the dominant organism in these BSCs and is clearly visible with the naked eye.

### 2.2 Sample collection

Sample collection was conducted in spring 2016 at both study sites. Samples were selected according to the dominant

occurrence of *N. commune* or *Z. ericetorum*. A green algal or cyanobacterial dominated BSC was defined as covering at least 50% of the soil surface in a 20 cm diameter petri dish. Once collected, the samples were first allowed to dry at room



temperature and then were kept frozen at -20 °C until used for measurements. Frozen storage is described as being suitable for long term storage of BSC components for experimental studies (Honegger, 2008).

## 2.3 CO₂ exchange

$CO_2$ exchange measurements were done according to Colesie et al. (2014). Before measurements, the intact BSC samples

underwent a reactivation procedure of two days exposure at 4 °C in the dark. Afterwards they were fixed in the gas exchange cuvette and sprayed with sterile, filtered water to activate their metabolism 24 hours prior to measurement. Ahead of the measurements, full water saturation was achieved by submerging the samples in water for ten minutes. Excessive water and droplets were carefully shaken from the sample before measurements.

CO2 gas exchange measurements were conducted under controlled laboratory conditions using a minicuvette system (GFS

3000, Walz Company, Effeltrich, Germany). The response of net photosynthesis (NP) and dark respiration (DR) to water content (WC) was determined for the cyanobacterial crust (n=six) and the green algal crust (n=4). Complete desiccation cycles (from the water saturated phase to the air-dried status) were measured under saturating light, ambient CO2 and a set of different temperatures (7, 12, 17, 25 °C). Within each crust type three sampling units were subject to measurements, which are referred to as organization levels. 1) Intact crust (named $BSC_{all}$): defined as the entire non-manipulated BSC

including attached soil with an unknown amount of heterotrophic organisms. 2) Separated organism (named $BSC_{dom}$): This sampling unit is the dominant organism of the crust, isolated from the from the soil, washed and dissected thoroughly. 3) Separated soil (named $BSC_{soil}$): This sampling unit contains the soil and all other microscopic organisms, except for the dominant one which was removed for type 2.

The following terminology will henceforth be used for the cyanocrust: $C\text{-}BSC_{all}$; $C\text{-}BSC_{dom}$ and $C\text{-}BSC_{soil}$; for the green algal

crust: $G\text{-}BSC_{all}$; $G\text{-}BSC_{dom}$ and $G\text{-}BSC_{soil}$.

Samples were weighed between each measurement cycle and the WC was then calculated as mm precipitation equivalent. Sample dry weight was determined after 3 days in a drying oven (Heraeus Instruments T6P, Thermo Fischer Scientific Inc.) at 60 °C. To obtain the net photosynthetic response to light, fully hydrated samples (n = three for each site) were exposed to stepwise increasing light from 0 to 2000 µmol photons $m^{-2}$ $s^{-1}$ (0, 25, 50, 100, 300, 500, 1000, 1500, 1750, 2000 photons $m^{-2}$

$s^{-1}$) at optimal temperature and ambient CO2 concentrations. The light cycle (about 40 min duration) was repeated until the samples were completely dry (after 6 – 7 h). Light saturation was defined as the photosynthetic photon flux density at 90% of maximum NP. The CO2 exchange of the samples was related to soil crust surface and chlorophyll content, the latter determined after Ronen and Galun, (1984).

To determine a possible effect of abiotic CO2 release from BSC attached soil, soil was measured before and after

autoclaving (see supplemental material Table S6).

**2.4 Species identification**

*N. commune* and *Z. ericetorum* were studied using a light microscope (Axiokop, Zeiss, Germany) and identified using Geitler, (1932) and Ettl & Gärtner, (1995). Additionally, other photoautotrophic cells from one gram of soil material were identified and counted, to get an overview of the photosynthetic active organisms that were present in the soil after the most abundant organism of the BSC had been removed.

**2.5 Data analysis**

To test for differences between mean values of cardinal points for photosynthesis (maximum NP and DR at the same temperatures, optimal water content range and water compensation points) for complete BSCs, soil alone and isolated organism, light compensation points and light saturation levels a multifactorial one-way analysis of variance (Statistica 10, Stat soft), with a Tukey post-hoc test were applied. Prior to the analysis, all data were checked for Gaussian distribution and homogeneity of variance and successfully log transformed if they did not fit these criteria. $BSC_{all}$, $BSC_{dom}$ and $BSC_{soil}$ were always the explanatory variables, while light compensation, light saturation, optimum water content, water compensation point, maximum net photosynthesis and maximum dark respiration were dependent variables. Paired t-tests were applied to detect differences in total NP rates of $BSC_{all}$ and NP rates of $BSC_{dom}$ and $BSC_{soil}$ taken together. The optimum water content was compared bystatistically testing if the upper and lower limits between the both $BSC_{dom}$, $BSC_{soil}$ or $BSC_{all}$ differ.

**3 Results**

**3.1 Gas exchange**

**3.1.1 Light dependent photosynthetic response**

Maximum photosynthetic rates per area were reached at 2000 µmol photons $m^{-2}$ $s^{-1}$ for both, C-$BSC_{dom}$ and G-$BSC_{dom}$ (Fig. 2a and 2b). Both organisms did not show photoinhibition even at the highest light intensities applied (2000 µmol photons $m^{-2}$ $s^{-1}$).

The light compensation point of photosynthesis was almost twice as much for G-$BSC_{all}$ (254 ± 53 µmol photons $m^{-2}$ $s^{-1}$) compared to C-$BSC_{all}$ (151 ± 25 µmol photons $m^{-2}$ $s^{-1}$). However, there was no statistical support for the difference between the organisms (grouped t-test: t-value=2.58; df=4; p=0.06), mostly owed to the high standard deviations caused by high sample variation.

Light saturation of NP (90% of maximal) was significantly different for both types and reached 985 ± 31 µmol photons $m^{-2}$ $s^{-1}$ for C-$BSC_{all}$ and 1260 ± 53 µmol photons $m^{-2}$ $s^{-1}$ for G-$BSC_{all}$ (grouped t-test: t-value=7.75; df=4; p=0.002).



### 3.1.2 Net photosynthesis

For *N. commune* (Fig. 3a) the organization level had a significant effect on the NP performance (F=38.06; p=0.000). NP was up to 4.3 times higher in C-BSC$_{dom}$ than in C-BSC$_{all}$ (Tukey post-hoc test: p= 0.000), whereas there was no difference between C-BSC$_{all}$ and C-BSC$_{soil}$. NP at 25 °C is significantly higher than at 7 °C and 12 °C for all, C-BSC$_{all}$, C-BSC$_{dom}$ and

C-BSC$_{soil}$ (F=9.41; p=0.000; Tukey post-hoc test: p 7 °C=0.000; p 12 °C=0.000).

In *Z. ericetorum* (Fig. 3b) NP also changed with organization level (F=53.61; p=0.000). There was a significant difference between G-BSC$_{soil}$ and G-BSC$_{all}$ (Tukey post-hoc test: p=0.03), and G-BSC$_{dom}$ NP differed from both (Tukey post-hoc test: p G-BSC$_{all}$=0.000; p G-BSC$_{soil}$=0.000). NP of an G-BSCd$_{om}$ was 5.5 times higher than in G-BSC$_{all}$. No temperature dependency of NP is visible for either the G-BSC$_{all}$, G-BSC$_{dom}$ or G-BSC$_{soil}$.

Effects of organization level and temperature on DR can be seen in Fig. S1 and Tables S7-S9 in supplemental material.

### 3.1.3 Water dependent photosynthetic response

The response to changing water contents in both BSCs and their separate components are shown in two exemplifying graphs (*N. commune* Fig. 4a and *Z. ericetorum* Fig. 4b). Neither the water-response curve of G-BSC$_{dom}$, nor C-BSC$_{dom}$, resembled

the response of BSC$_{all}$. The NP in C-BSC$_{dom}$ was three times higher compared to C-BSC$_{all}$, and doubled in G-BSC$_{dom}$ (for statistical analyses see Table S1-3 supplemental material). Respiration in C-BSC$_{soil}$ was strikingly small in an *N. commune* dominated system and did not resemble the respiratory response in C-BSC$_{all}$ or the C-BSCd$_{om}$, whereas respiration was pronounced in a *Z. ericetorum* dominated G-BSC$_{all}$. Respiration of G-BSC$_{soil}$, G-BSC$_{all}$ and G-BSC$_{dom}$ all seemed to follow the same pattern in the green algal dominated BSC, although the values differed slightly between all three, with highest rates

in G-BSC$_{soil}$. A general observable pattern was that NP of BSC$_{soil}$ and BSC$_{dom}$ combined never equalled NP of BSC$_{all}$ in both BSCs (paired t-test; results for *N. commune*-crust: t-value=-6.43; df=23; p=0.000; *Z. ericetorum*-crust: t-value=-5.05; df=15; p=0.000).

C-BSC$_{dom}$ showed a very slight depression in net photosynthesis at high water contents. Also, only a small depression in the C-BSC$_{all}$ was visible. In G-BSC$_{dom}$ and in G-BSC$_{all}$ a clear depression was expressed between 80–100% normalized water

content.

The optimum water content had a broader range for C-BSC$_{all}$ (0.63 to 1.20 mm) compared to C-BSC$_{dom}$ (0.18 to 0.26 mm). For G-BSC$_{all}$ it was between 0.88 and 1.26 and for G-BSC$_{dom}$ between 0.25 and 0.27, so the values between both crust types and their isolated organisms was close. Both BSC$_{all}$ optimum water content ranges were significantly higher than the corresponding BSC$_{dom}$ optimum water content ranges (multifactorial ANOVA *N. commune*: df=2; F=15.24; p=0.000;

multifactorial ANOVA *Z. ericetorum*: df=2; F=30.08; p=0.000). The optimal water content differed heavily with organization level and was broadest in BSC$_{dom}$ for both BSC-types (Fig. 5; *N. commune*: F=43.20; p=0.000; *Z. ericetorum*: F=66.28; p=0.000). This pattern is expected, as extracellular polymeric substances have a different water holding capacity

than soil. Nonetheless, both BSC$_{all}$ and their respective BSC$_{dom}$ showed a broad optimum water content range (Fig. 5), compared to soil alone. Both limits of BSC$_{dom}$ are not statistically different, as also in BSC$_{all}$ (for statistical results see Table S4 & S5 in supplemental material)

## 3.2 Net photosynthesis per area vs. net photosynthesis per chlorophyll content

A significant difference between NP calculated per area and NP per chlorophyll content became obvious after comparing the results. Besides having an up to 181 times higher chlorophyll content per area (Table. 1), net photosynthesis of G-BSC$_{dom}$ was always up to two magnitudes lower than in C-BSC$_{dom}$. To investigate whether this difference was due to actual eco-physiological differences or because of the reference value used, an "effect size" was calculated. The ratio between NP at a certain temperature between C-BSC$_{dom}$ and G-BSC$_{dom}$, based on area was calculated. The same was done for NP rates based on chlorophyll content and besides the single organism also for BSC$_{all}$. The difference of chlorophyll based NP between C-BSC$_{dom}$ and G-BSC$_{dom}$ and in BSC$_{all}$, was always higher (up to 42 times in BSC$_{all}$ 25 °C), than the difference in area based NP (grouped t-test; t-value=-2.167; df=14; p=0.05; Fig. 6), although the ratio should be similar, if both reference values (area and chlorophyll content) were equally suited for gas exchange measurements in terrestrial cyanobacteria and green algae crusts. A difference with temperature might be explained by the dependency of net photosynthetic enzyme activity on temperatures.

## 3.3 Photosynthetic active and heterotrophic organisms in soil

The number of cells of photosynthetic active organisms in soil samples was similar for both study sites ($26 \pm 3$ organisms per gram soil for MH and $19 \pm 2$ for PL). The abiotic gas release, measured after autoclaving the soil, was about 20 % of the gas exchange before autoclaving (Table 6 suppl. Material).

## 4 Discussion

In this study, we received profound evidence that early successional BSCs expose a considerable physiological plasticity which in turn might be responsible for their pronounced stress tolerance and finally their success in heavily disturbed habitats. This will be discussed here in detail. As light is one of the major drivers for photosynthetic rates in BSCs, the cardinal points in response to light were examined. For BSCs the light level compensating respiration normally is between 60 to 100 photons m-2 s$^{-1}$ (Green & Proctor, 2016), which indicates comparability to classical "sun plant" features. Our results support these findings and the variation within our measurements, indicated by high standard deviations, reflects a very high sampling site internal variation. This may be a result of spatially small-scale shading of, e.g. *Calluna vulgaris* in the Mehlinger heathland, or higher plants at the border of Parking Lot. As high light adapted organisms, light saturation



points of BSC organisms are almost always at, or above, 700 µmol photons $m^{-2}$ $s^{-1}$ (Green & Proctor, 2016). This is well supported by our data with LSP consistently being over 900 µmol photons $m^{-2}$ $s^{-1}$ for both, C-BSC$_{all}$ and G-BSC$_{all}$.

The second major driver for photosynthetic responses in BSCs is water, thus the photosynthetic cardinal points in response to water content were examined. The lack of differences in the optimal water content between C-BSC$_{dom}$ and G-BSC$_{dom}$, and none between the green algal and cyanobacterial BSC$_{all}$, indicates a high physiological flexibility of the organisms to water gain. Explaining this variability of eco-physiological response to water content are the relatively high standard deviations and the broad range of the optimal water contents for the dominant organisms, compared to other species (e.g. 0.25 – 0.35 mm in Fulgensia fulgens (Lange et al., 1998)). Optimal water content of a BSC in a temperate habitat (Homburg; 3.32 mm; Europe, Germany; Colesie et al., 2014b) is 2 mm higher than in our study (1.30 – 1.46 mm), probably owed to the low biomass in the investigated early successional habitats in this study. Colesie et al., (2014b) proposed that biomass is the driving factor shaping the optimal water content within a system, with low biomass needing less water to achieve maximum net photosynthesis. A general difference between BSC$_{all}$ and BSC$_{dom}$ concerning optimal water content is likely owed to the different water holding capacities of the soil. Even at high water contents, C-BSC$_{dom}$ only showed a slight depression of NP (Fig. 4). Depression in G-BSC$_{dom}$ occurred relatively late. Considering that these crusts live in a temperate climate, where precipitation is high enough to support forest vegetation, a depression of NP at high water contents might be a disadvantage, although a depression is common in other BSC organisms (e.g. several lichens; Lange et al., 1997). Additionally, *N. commune* can gain up to 20–30 times its dry weight in water (Satho et al., 2002; Shaw et al., 2003), which is stored mostly in exopolymeric substances (EPS) (Tomaselli & Giovannetti, 1993; Kovacik, 2000). A depression of NP at a high water content while simultaneously being exposed to high water contents for a long time period would be disadvantageous. *Z. ericetorum* only shows a slight difference in morphology when dried (Fritsch, 1916; Holzinger et al., 2010), which means that it does not store as much water as *N. commune*, and is not as often as strongly exposed to high water contents around the cell. Therefore, it might have a depression of net photosynthesis at high water contents. Additionally, a carbon concentrating mechanism (CCM) was detected in *N. commune* (see supplemental material Fig. S2), but not in *Z. ericetorum*. Through this mechanism, the internal $CO_2$ partial pressure around the carboxylating enzymes is increased, which improves photosynthetic efficiency. Simultaneously, CCMs decrease the $CO_2$ concentration around the photosynthesizing cells. Through this increasing diffusion gradient for $CO_2$ transport the adverse effects of suprasaturation with water can be overcome (Lange, 2001).

The third major driver of photosynthesis in BSCs is temperature. Temperature showed different effects on net photosynthesis rates in both organism groups. In C-BSC$_{all}$ NP was only higher at 17 °C and 25 °C compared to lower temperatures (7 °C and 12 °C), whereas in G-BSC$_{all}$ no difference was observable at any temperature. This indicates a high variability in temperature tolerance, which was also noticed by Borstoff et al. (2005), who could not pinpoint an exact temperature optimum for a BSC during field studies in western Mojave Desert. BSC components are not necessarily adapted to high temperatures. Although soil surfaces at which BSCs live can reach up to 50 °C on a normal summer day (Hoppert et al., 2004; Pentecost & Whitton, 2012), the organisms are already dry and inactive when these temperatures are reached (Raggio



et al. 2017). Presumably this is the case for G-BSC$_{all}$, but not likely for the C-BSC$_{all}$ in this study, due to the high water holding capacities of Nostoc thalli, therefore activity periods during daytime are prolonged. This may coincide with a temperature increase, as it has been reported for gelatinous, slow drying lichens (e.g. Collema tenax; Lange, 2001). In Antarctica, Novis et al., (2007) conducted a study on the importance of *N. commune* in antarctic habitats, it was observed

that this organism tends to have its highest NP rates at unusually high temperatures (20.5 °C). However, the results of Novis et al., (2007) should be treated with caution, because it is very likely that organisms corresponding to the description of *N. commune* from polar and temperate regions may be genetically distinct (Novis & Smissen, 2006) and therefore should not be compared so readily. Nevertheless, in general cyanobacteria are better adapted to high temperatures than green algae are (Castenholz & Waterbury, 1989; Lange et al., 1998).

Summarized, both crust types were largely unaffected by changes in water content and had high light compensation and exceptionally high light saturation points. Net photosynthesis of G-BSC$_{all}$ was unaffected by increasing temperatures, while C-BSC$_{all}$ only had increased NP rates with higher temperatures. Both fit well within known eco-physiological patterns of different BSC components (see Lange & Belnap, 2001), and show a lower net photosynthesis compared to BSC components of later successional stages (see e.g. Lange, 2001). The high physiological variability in photosynthetic cardinal points places

*N. commune* and *Z. ericetorum* crusts as similar to other BSC contributors, such as lichens and bryophytes. This characterizes these organisms as stress tolerators with typical patterns of Grimes CSR-model (Grime & Pierce, 2012). Life history traits of stress tolerators include slow growth rates, high rates of nutrient retention, low phenotypic plasticity and a response to environmental stresses through physiological variability (Grime & Pierce, 2012).

In addition to this eco-physiological description, methodological procedures are discussed for future investigations on BSCs.

First, an unavoidable complication in gas exchange measurements of BSCs is to take abiotic release of CO2 into account (Inglima et al., 2009; Weber et al., 2012), aside from biotic respiration of BSC organisms, phototrophic and heterotrophic. The CO2 release of alkaline and saline substrates can even exceed those of organic activity (Xie et al., 2009; Shanhun et al., 2012; Ma et al., 2013; Sancho et al., 2016). The abiotic release in this study accounted for only 20 % and 26 % (MH and PL, respectively) of total soil respiration rate, which is close to what Weber et al., (2012) measured in the Succulent Karoo or

Shanhun et al., (2013) with 25% in Antarctica.

Secondly, besides an obvious effect of organization level on NP and DR rates (see Fig. 3, and supplemental material Fig. S1), the water response curves BSC$_{all}$ and BSC$_{dom}$ alone differed. Although the courses of their curves were similar, net photosynthesis rates never were equally high for BSC$_{all}$ and BSC$_{dom}$ (Fig. 4a and b). Adding NP rates of BSC$_{soil}$ to BSC$_{dom}$ did not equal the values and NP was always higher in BSC$_{dom}$ compared to BSC$_{all}$ in both crust types. This fact confirms the

view of BSCs as communities or systems rather than random accumulations of organisms and strengthens the need for long term field monitoring studies of an intact BSC. Causes that might have led to an altered NP rate could be self-shading or water logging. An organism laying on top of an unaltered BSC has a species specific structure (e.g. lobes of *N. commune* over one another or the spherical structure of *Z. ericetorum* algal mats) that will be changed and rearranged while being washed and placed separately in the cuvette of the GFS 3000. Through this thallus exposure not only internal diffusion



resistance is decreased, but also the self-shading protective mechanisms in those organisms are no longer functional. Aeroterrestrial filamentous green algae, such as Klebsormidium crenulatum form multilayers mat like structures on top of or interwoven with the upper millimeters of soil (Karsten et al., 2010b), contributing to self-shading of individual filaments inside a population. The same is visible in *Z. ericetorum*. While rearranging the organisms the protective layer of dead or

highly melanosed cells on top of the crust is either removed or shifted and unprotected but highly active cells are now subjected to high light intensities. Although high light intensities will cause damage, e.g. through reactive oxygen species production, the NP response itself is higher than before. Overlapping lobes in N commune also may cause reduced NP rates in an intact BSC system. Bowker et al., (2002) hypothesized that sunscreen pigments produced by Nostoc protect other, less pigmented taxa, which does not exclude a self-shading protective mechanism in lower lobes of the same species. Recent data

suggests that shading effects due to such three dimensional spatial arrangements may be more important than previously assumed (Karsten & Rindi, 2010). Another possibility that might lead to decreased NP in a complete BSC system is water logging between filaments of the organisms (Garcia-Pichel & Belnap, 2001). The diffusion resistance of $CO_2$ in water is higher than in air (Cowan et al., 1992), and will reduce NP rates. A water suprasaturated system, as it can be found in the closely growing mats of *Z. ericetorum* or *N. commune*, could have a disadvantage against a widely spread organism surface

in the cuvette, that is aerated all around.

Thirdly, we propose a methodological consideration concerning reference values of photosynthesis in gas measurements in general. To compare cardinal points of photosynthesis between organisms or samples of the same species it is necessary to base the photosynthetic values on either dry weight, area or chlorophyll content, which are the most common reference values in literature. In Fig. 6 it becomes apparent that the ratio of NP per area and NP per chlorophyll content in C-BSC$_{dom}$

and G-BSC$_{dom}$ differ heavily, although the ratio should be similar if both reference values were equally suited for gas exchange measurements in those organisms. This difference is most probably owed to the completely different composition of photoactive pigments in the two organism groups. Terrestrial green algae possess chlorophyll a and b, whereas cyanobacteria only possess chlorophyll a, which is considered in the calculation of chlorophyll content. But cyanobacteria also possess phycobilisomes and super antennae allowing capture of photosynthetically active radiation from low photon

flux densities and the green part of the spectrum (Lüttge, 2011), which are not considered in calculations of this reference value. A much more suitable value than chlorophyll content is therefore area, which has been used in this study. Nonetheless, this value is not perfect as there are probably many dead cells still bound to the EPS of living cells that distort area measurements, and of course it is not useful if fruticose lichens are investigated. Additionally, the arrangement of the BSC does have an influence on the photosynthetic signal, considering that *N. commune* is almost flat while *Z. ericetorum* has

a more spherical structure as the BSC gets older. It is therefore of immense importance to decide which reference value should be used when comparing the eco-physiological response of BSC systems.



## 5 Conclusion

In this study, we present the first detailed eco-physiological dataset describing photosynthetic performance of early successional BSCs. A relative temperature independence of NP, as well as late or no water depression and an adaptation to high light intensities was demonstrated for both early successional BSCs and their separated dominant organisms. This physiological plasticity indicates strong stress tolerance in both organism groups and might be the reason for their success in heavily disturbed areas. The results can be incorporated into global scale carbon cycle models. Additionally, this study emphasizes the importance of measuring a complete BSC rather than their single components, as not only taxonomic composition but also spatial arrangement seems to be an important factor shaping photosynthetic response in BSC systems. The methodological approach demonstrated that a comparison of photosynthetic values in cyanobacteria and green algae should be based on area rather than chlorophyll content.

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

**Tables**

**Table 1: Chlorophyll content per area in *N. commune* and *Z. ericetorum* samples without soil.**

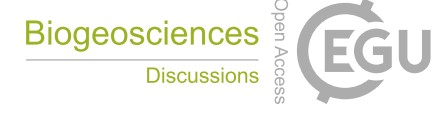

| Sample | Chlorophyll content per insolated area [mg/cm²] |
|---|---|
| *N. commune* S1 | 0.23 |
| *N. commune* S2 | 0.10 |
| *N. commune* S3 | 0.49 |
| *N. commune* S4 | 1.48 |
| *N. commune* S5 | 0.17 |
| *N. commune* S6 | 0.36 |
| | |
| *Z. ericetorum* S1 | 18.13 |
| *Z. ericetorum* S2 | 1.11 |
| *Z. ericetorum* S3 | 15.79 |
| *Z. ericetorum* S4 | 2.68 |

## Figures

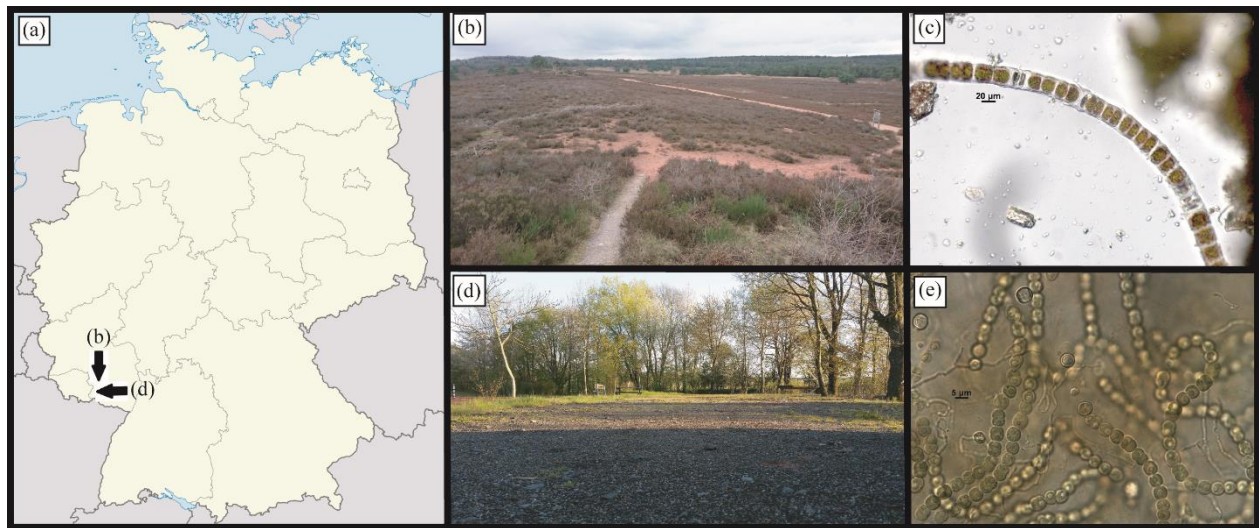

**Figure 1: Organisms and map of the study site in Germany, Rhineland–Palatinate (a): Top pictures are from study site 1,**
5 **Mehlinger Heath viewed from the look-out (b) and microscopic images of the crust dominating organism *Zygogonium ericetorum***
**(c); bottom depicts study site 2, a parking lot at an equestrian farm near Zweibrücken (d) and microscopic images of the crust**
**dominating organism *Nostoc commune* (e).**

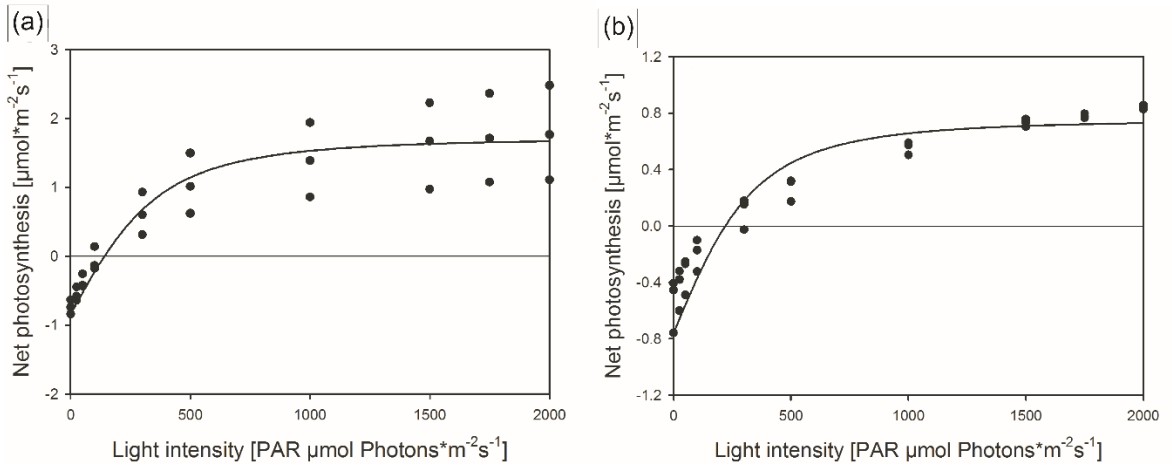

**Figure 2: Light dependent photosynthetic response curves of (a)** *Nostoc commune* **and (b)** *Zygogonium ericetorum***. Net photosynthesis per area is plotted against light intensity as PAR [µmol photons m$^{-2}$ s$^{-1}$] with n=3 samples.**





**Figure 3: Temperature dependent net photosynthesis per area in (a)** *N. commune* **and (b)** *Z ericetorum* **dominated BSCs, separated dominant organism and separate soil. Capital letters indicate significant differences in organization level between BSC, organism and soil; lower case letters compare temperature differences of one of the groups only. Sample size was n=6 for** *N. commune***; n=4 for** *Z. ericetorum***.**



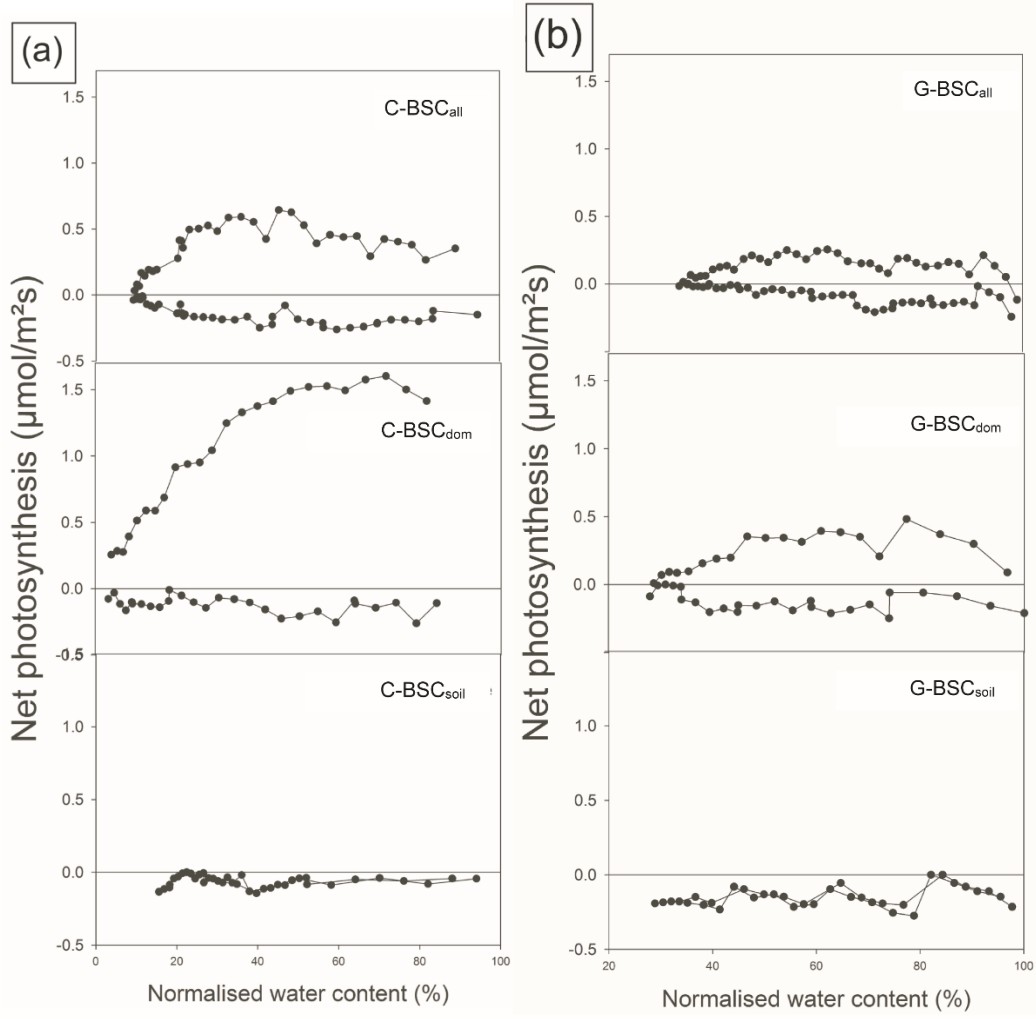

**Figure 4: Differences in water content dependent photosynthetic response in an intact BSC, the isolated dominant organism and in soil. Net photosynthesis is plotted against normalized water content for (a) *N. commune* and (b) *Z. ericetorum*, both at 12 °C**



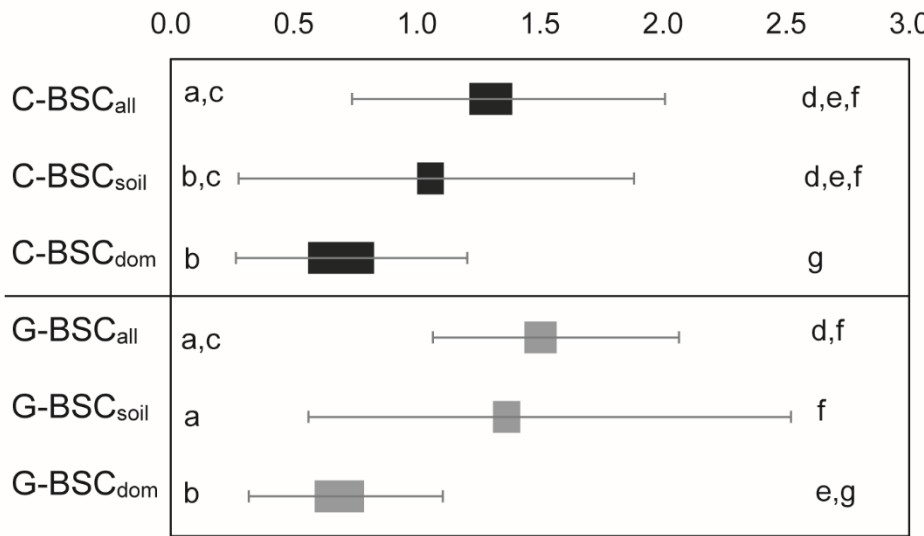

**Figure 5: Optimum water content range, in mm precipitation, of a cyanobacteria dominated BSC (upper three; black bars) and a green algal dominated BSC (lower three; grey bars) and their respective separated components. Sample size is always n=24.**

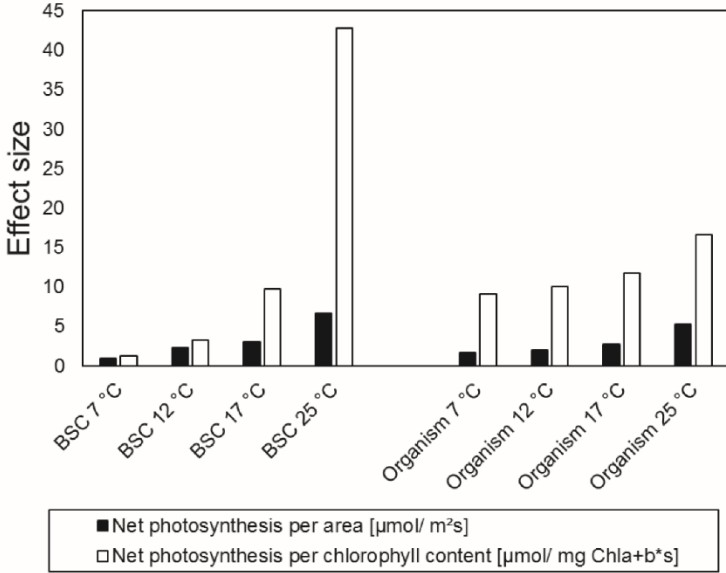

**Figure 6: Difference in the ratio of net photosynthesis based on either area or chlorophyll content between _N. commune_ and _Z. ericetorum_ and their respective BSCs, dependent on temperature.**