# Peer review of "Eco-physiological characterization of early successional biological soil crusts in heavily human impacted areas"

_Biogeosciences, 2017_

## Referee Comment (RC1) · T. Fischer (Referee) · 12 Oct 2017

The particular value of the study is the comparison of the eco-physiological performance between a cyanobacterial and a green-algal biocrust from temperate habitats, which are somewhat underrepresented in the biocrust related literature. I recommend publication of the manuscript after minor revision.

Minor remarks

Remark #1: Figure 4: I guess the upper line in each graph is under light, and the lower

line in the dark? What was the PPFD?

Remark #2: p. 7 l. 26 and Fig. 5: Water contents are given in mm here, but as normalised water content in the rest of the manuscript. I think the paper would benefit from providing some information on how many mm were 100% for each BSC type. For soils, water content expressed as mm links with volumetric water content (or water potential) through soil texture, depth and humus content, which are essential to relate to each other optimum moisture ranges for BSC_all, BSC_dom and BSC_soil. While, for example, the optimal ranges for G-BSC_dom and C-BSC_dom are similar, the difference between G-BSC_dom and G-BSC_soil is larger than the respective difference for the cyanobacterial crust: This could mean that the amount of fine particles, or sampling depth, or soil C, or all together, were greater for the Zygogonium crust. The authors are aware of that point (p. 9 l. 12-13): "A general difference between BSC_all and BSC_dom concerning optimal water content is likely owed to the different water holding capacities of the soil."

Remark #3: p. 8 l. 19-20: High abiotic $CO_2$-release may point to carbonates being present in the soil solution and to high pH. The authors discuss that issue on p. 10 l. 20 ff.

Remarks #2 and #3 let me recommend to provide some information on soil texture class, pH, organic C content and sampling depth for each site in the M&M section.

Remark #4: p. 10 l. 1-2: The authors state a higher water holding capacity (WHC) of the Nostoc crust than the Zygogonium crust and attribute this to exopolysaccharides (EPS), which is in full agreement with the literature. However, apart from its lower NP performance, the Zygogonium crust had higher amounts of chlorophyll (Table 1), which traditionally is interpreted as a biomass equivalent. Is it possible that high Zygogonium biomass compensates for high WHC of the EPS of Nostoc? I think that the statement of higher WHC of the Nostoc crust could be substantiated by some experimental data, or, for example, from presenting some close-up photographs of the crusts to get a visual

impression of crust development.

---

## Referee Comment (RC2) · Anonymous Referee #2 · 16 Oct 2017

Dear Editor, dear authors: Please find here my review concerning the paper "Eco-physiological characterization of early successional biological soil crusts in heavily human impacted areas – Implications for conservation and succession" by the authors Michelle Szyja, Burkhard Büdel, and Claudia Colesie. I hope that all of you find comments made useful and that they can improve the final manuscript at some extent.

The manuscript is an interest piece of research with a main input to the general state of the art, which is, under my point of view, the ecophysiological characterization of early successional stages of biocrusts. These types of biocrusts are not often analyzed in

the literature, more focused in later stages of development. The analysis is pertinent in the sense that desertification seems to be a real threat worldwide, and that climatic conditions predicted for the future in several parts of the planet are going to change regarding water availability and distribution. Thus, early successional stages of biocrusts are not only important due to the fact of being pioneer colonizers of barren habitats, but also because climatic predictions point to them as possible dominant organisms in areas under strong hydric stress, where later successional stages would not succeed. Up to the authors to use this concept in the introduction in order to remark even more the importance of their study

Besides this, the manuscript offers, as authors underline, relevant C fluxes data set of not often analyzed biocrusts, which are valuable data for modelling attempts regarding the relevance of these organisms in C cycling and the impact that future environmental changes will have over this. Some of the current models available lacks of proper reliability and only by doing direct measurements of different biocrust types at different parts of the world will be possible to overcome these problems

Finally, authors provide quantification about inorganic fluxes of carbon released in the samples studied, information that is always useful to know before designing an experiment about gas exchange with samples with soil attached. The more we know about this, the better and accurate data sets we will produce.

Due to these points, I would recommend publication of the paper if the authors are able of changing and/or explaining some points that I do not see clear or that, at least, I did not understand properly.

In detail:

INTRODUCTION

Page 4 lines 4-5: Please clarify this sentence. I think that authors want to say here that depending of the treatment made to the sample (sample with soil, without it, or bare

soil) a different response will be found in the gas exchange experiments. But I do not understand the sentence: "We expect that the position and arrangement of the sample inside the measurement system, here a cuvette, will influence the photosynthetic values". Are the authors analyzing, at some point, how the position of the sample inside the cuvette is influencing gas exchange measurements?

I think that the sentence is confusing and is not a good choice to close a, on the other hand, well developed introduction

MATERIAL AND METHODS

P5, L13: Could authors provide some info about why was this set of temperatures chosen for the experiment?

P6, L9: I do not see clear how a one way anova can be, at the same time, a multifactorial anova. To my understanding, the authors are using a one way anova with type of crust being the factor (meaning that only on efactor is being analyzed), and each of the dependent variables analyzed at each moment (NP, DR, WC......) being the variable. Is this correct? Probably just a matter of terminology but I see it a bit confussing as written now

P6, L15: A space is needed in "bystatistically". Besides, which methodology was used to compare these limits?

RESULTS

P7 L23-25. After having a look to Fig. 4 I agree with what is written here, but I think that is falling in contradiction with what is written in the abstract about the issue: "and low or no depression in carbon uptake at water suprasaturation" (abstract L18). I think that the text in the abstract regarding this issue should be changed to fit more accurately what is written in results

P7 L27-28: I think that what authors want to underline here is that C-BSC and G-BSC water content values are close between them both situations, "all" samples and "dom"

samples. But as it is written now it seems that, for example, for C-BSC "all" and "dom" values are similar between them, which does not seem to be correct. Just a small correction would solve the possible confusion.

FIGURES

Fig. 2. I think that both sub-graphs should be scaled equally at the Y axis in order to compare gas exchange rates between C and G crust types easily

Fig. 4. Please indicate in the figure legend the amount of light used for the experiment

Fig. 6. This figure is hard to follow for me. I think that the variable "effect size" is a ratio between C and G crust types calculated for "dom" and "all" samples and based on area of each sample and chlorophyll content, but I do not understand why such ratio is called "effect size". Could authors please provide more explanations about this graph? I do not understand either that bump of the effect size at 25 °C for chlorophyll based net photosynthesis. I have read in different parts of the text that authors consider that net photosynthesis has not a statically significant drift with temperature on an area basis, at least for the green algae crust. Does this graph mean that temperature has a significant effect over photosynthesis on a chlorophyll basis but not on area basis? Besides, the figure is supposed to show differences in the effect size for both N. commune crusts and Z. ericetorum, but I do not see clearly which is which in the graph.

DISCUSSION

P9 L13-22: Authors discuss in this paragraph about the differences in depression of net photosynthesis at high water content between C "dom" crusts and G "domÂĺ crusts, explaining ecologically why makes sense the fact of not finding this depression in Nostoc (C) and finding it in Zygogonium (G). After having a look at figure 4, it seems to me that there are more measured points at high water content (over 80% of maximum water content) in Zygogonium than in Nostoc (I mean, for CBSC dom it seems that there is a gap between 80% and 100% of water content). Any explanation for this?

Could this affect the ecological interpretation of the depression of net photosynthesis at high water contents or authors are using other indicators to analyze this issue?

P9 L23-27: I have gone to the supplement figure S2 in order to try to follow the detection of the CCM mechanism and its relationship with depression of photosynthesis at high water contents. This is something quite interesting physiologically under my pointy of view that deserves more research efforts in the literature. I have seen that authors propose (correct me if I am wrong) that the fast changes in differential $CO_2$ response in the gas analyzer after light changes supports the existence of the CCM in Nostoc, and that this was not found in Zygogonium. Do you mean that the response of Zygogonium after light changes was different or somehow slower that in Nostoc? Is there any support in the literature for this pattern? (I mean presence or absence of CCMs in cyanobacterias Vs green algae)

P10 L1-2: I have been following with interest the lack of optimum temperature for net photosynthesis in the green algal crusts because it was something initially unexpected to me. What I see in relation to this in Fig. 3 regarding C and G "dom" subgraphs, is that Nostoc follows a pattern of raised net photosynthesis with temperature through all the temperature range and that Zygogonium shows a raise up to 17°C and a decrease at 22°C (but 22 showed highest photosynthesis compared with 12°C). I know that authors are supporting their idea of lack of temperature optima in the stats, which I think that is right and interesting, but after looking the graphs it seems to me that it could be perfectly said that Nostoc dom has a temperature optima at 25°C and Zygogonium at 20°C.

If Zygogonium is less adapted to long activity periods than Nostoc, I would expect a concentration of metabolic activity during softer environmental conditions, and this should shift temperature optimals to lower values rather that erase the concept of optimum temperature for net photosynthesis. On the other hand, author's statement of lack of temperature optima in the green algae is supported with the graph 3b for G-BSC all, where the link between net photosynthesis and temperature is clearly erratic

and defined by a lack of pattern. I just would like to know author's opinion about this, because their approach to T optima concept based in stats is absolutely right to me.

And this is a different issue, but it is surprising to me the lack of statistical differences in Nostoc between C-BSCall and C-BSCsoil net photosynthesis. It means that the photosynthetic cyanobacteria layer of the soil is not creating any relevant C input compared with bare soil. Interpretations for this behaviour?
* * *

---

## Referee Comment (RC3) · Anonymous Referee #3 · 8 Nov 2017

Comments to the Authors

The paper of Szyja et al. aims to characterize ecophysiologically early successional biological soil crusts in heavily human impacted areas. For achieving this they choose two locations with a different type of BSC: one dominated by a cyanobacteria and the other by a green alga. Overall, I found the paper representing an interesting contribution to scientific knowledge of BSC ecophysiology because: 1- there are at present not many data available about ecophysiology performance of these type of BSCs and 2- The comparison of the response between bare soil, intact BSC and isolated component is novel and very interesting. Nevertheless, I found some important problems as how the work is presented. The main problems are in the methodology where the experimental design (mainly number of replicates in each experiment) is not clear and in the results, where some of the figures are quite confusing. The question about whether the NP rates should be expressed on a chlorophyll or surface basis is not relevant here and, obviously, will differ if comparisons are made between cyanobacteria and green algae. In my opinion the number of references (85) exceeds the needs of the paper. Beside some minor/typographic errors (i.e. check subscript in $CO_2$ throughout the text), in general, the paper is well-structured, the discussion is good and conclusions clear but it needs to show results in a way that they appear more conclusive.

In conclusion, I find the paper interesting and scientifically sound but taking into account the amount of data and how they are presented I don't think it reach the standards of BG. I have some comments and suggestions that I think will improve the paper.

Major and minor comments

Title

- I suggest removing the second part of the title (implications for conservation and succession) as it does not reflect the content of the paper.

Abstract

- There is no reference in the abstract to one of the main points in the work that is the differences found between response of intact BSCs and of it isolated dominant components.

- Page 1. Line 20. I suggest to remove the sentence beginning "Nevertheless, a major. . ." See comment above.

Introduction

- Page 1. Line 29. Please rewrite the sentence "Investigations. . ." As it is now is

contradictory. Are there abundant or few investigations in cyanobacteria?

- Page 2. Lines 5 to 20. In my opinion the concept of arrested succession should be introduced at the beginning so it is clearer for the reader.

- Page 3. Line 9. Reference Reisser et al. 2007 is not in the list.

- Line 23. I suggest to change "or" for "and a"

- Lines 25-26. Were these "in situ" measurements carried out? I think it would be better say "would allow"

- Line 32. Colesie et al. 2014b not in the list. "Higher" than what?

- Page 4. Line 4. The sentence is confusing and I think is not relevant here. I assume that when authors refers to system they refer to BSC and not to the measurement systems. The treatment or position in the cuvette is another question. Of course there will be variability between samples, but here the comparison is between isolated individuals (green algae or cyanobacteria), soil biocrust and soil. I suggest removing this sentence. Material and methods

- Page 5. Line 2. Check reference Honegger 2008. Is 2003 and also it refers to green algae photobiont but not to cyanobacteria.

- Line 11. n=6. It is not clear to me how the sampling or subsampling was made. From each 6 of C-BSC and 4 of G-BSC you take 3 subsamples?

- Line 12. First, you need to indicate how the saturation light was determined.

- Line 16. Delete "from the"

- Line 19. Should not be a new paragraph.

- Line 21. I understand that the weighing was during the dehydration cycle to have the full response, but not between them. Please explain this.

- Line 23. I suggest new paragraph. "To obtain the net response to light. . ." n=3. Are

the samples BSC or species individuals? It is not clear from the text and in Fig. 2 they appear as individual species measurements.

- Line 25. How the optimal temperature was obtain? Are there any regressions done for this? Data is not show. Please explain.

- Line 29. Should not be a new paragraph.

- Page 6. Line 10. Include "of the two types of BSC" after "levels". The analysis as it is explained is confusing as there were different number of samples and subsamples for the different experiments. For the drying curves there were 6 C-BSC and 4 G-BSC and from each of these all, dom and soil. But in the light curves there are only 3 C-BSC and 3 G-BSC without distinction of components. So, I understand that BSCall, BSCdom and BSCsoil cannot always be the explanatory variables. Results

- The adjustments of the curves in Fig. 2 doesn't look very good, especially that of Z. ericetorum, showing an increase in the response and no saturation following the points and not the line. Please check this. Also, how where the light parameters (compensation and saturation) calculated, from individual adjusted curves or from one curve? It should be explain in material and methods. There are no supplementary tables or graphs showing values of these parameters.

- Page 6. Line 4. From Figs 4a and b they don't contribute to NP response.

- Line 22. Here it is said G-BSCall and C-BSCall but not in Fig. 2. Please clarify. I suggest changing "almost twice as much" for "higher"

- Line 24. Why organisms? Is it not BSC? It is not reasonable that the difference in compensation point was twice as much but then there were no significant. As comment above please explain how this analysis was done.

- Line 26. The same discussion will apply for the saturation points. From Fig. 2 we can understand that there is no saturation at 2000 $\mu$mol (just a few lines before it is said that maximum NP rates were reached at 2000 $\mu$mol).

- Page 7. Line 2. Include "dominated BSC" after "commune". Refer here to Supplementary tables.

- Line 6. Include "dominated BSC" after "ericetorum". Refer here to Supplementary tables.

- Line 8. Delete "an"

- Water dependent photosynthetic response. In my opinion better than exemplifying graphs, average data of all replicates should be represented. Differences between just two samples are not relevant. Also, curves shown in Figures 4 are very difficult to understand as it is not normal the fluctuation around 80% water content. It must be an artifact that could be masked using averages. Also the water depression is not clear.

- Line 16. Change Table for Tables.

- Lines 26-28. Data shown in the text of ranges of optimal WC seem different from the ones in Fig. 5 (i.e. upper limit never coincident). Please check.

- Page 8. Line 5. I would rather delete this subsection as discussed above.

- Line 20. Table S6

Discussion

- Line 25. BSCs photosynthetic organisms

- Page 9. Line 5. Delete "none" and better G-BSCall and C-BSCall. What does it means physiological flexibility to water gain?

- Line 18. I suggest delete sentence beginning "A depression..." as it has already said before.

- Line 23. I don't see the detection of a CCM from Fig.S2.

Page 11. Lines 2 and 7. Species name in italics.

-Lines 16-26. In my opinion this question is not relevant as it is obvious.

Conclusions

Page 12. Line 3. The authors conclude that there is a relative temperature independence of NP but the results show significant differences in the response of NP to temperature.

- Line 5. In general, the question about physiological plasticity should be avoid because there are no experiments proving this.

- Line 6. To incorporate the results into global scale carbon cycle models, the work should better provide numerical data sets (i.e. tables).

References

There are too many for the paper. As mentioned above some cited literature in the text is not in the list. Please check references through the list.

Table 1. Following my suggestion about Chlorophyll question then this should not be included.

Figure 2. Legend.The second sentence is not necessary, just n=3.

Figure 3. Legend. What do you mean by . . .of one of the group only? Please indicate what vertical bars represent.

Figure 4. See comment above. Indicate PAR

Figure 5. This graph is very difficult to understand. See comments above. What does the letters mean? Why n=24 here?

Figure 6. As suggested above I would not include this graph.

[Figure]

---

## Author Comment (AC1) · 12 Dec 2017

Referee #1, Thomas Fischer The particular value of the study is the comparison of the eco-physiological performance between a cyanobacterial and a green-algal biocrust from temperate habitats, which are somewhat underrepresented in the biocrust related literature. I recommend publication of the manuscript after minor revision.

Minor remarks

Remark #1: Figure 4: I guess the upper line in each graph is under light, and the lower

line in the dark? What was the PPFD?

We have edited figure itself and the figure caption as follows, to clarify which line represents which physiological process. Also, we have added information about the light intensity (PPFD) applied during the measurements.

"Figure 4: Responses of net photosynthesis (dots) and dark respiration (triangles) to normalized water content for intact BSC, the isolated dominant organisms and in soil at 12°C. Measurements were taken at saturating light and a temperature of 12°C. (a) C-BSC (985 $\mu$mol photons m-2 s-1) and (b) G-BSC (1260 $\mu$mol photons m-2 s-1)."

Remark #2: p. 7 l. 26 and Fig. 5: Water contents are given in mm here, but as normalized water content in the rest of the manuscript. I think the paper would benefit from providing some information on how many mm were 100% for each BSC type. For soils, water content expressed as mm links with volumetric water content (or water potential) through soil texture, depth and humus content, which are essential to relate to each other optimum moisture ranges for BSC_all, BSC_dom and BSC_soil. While, for example, the optimal ranges for G-BSC_dom and C-BSC_dom are similar, the difference between G-BSC_dom and G-BSC_soil is larger than the respective difference for the cyanobacterial crust: This could mean that the amount of fine particles, or sampling depth, or soil C, or all together, were greater for the Zygogonium crust. The authors are aware of that point (p. 9 l. 12-13): "A general difference between BSC_all and BSC_dom concerning optimal water content is likely owed to the different water holding capacities of the soil."

Remark #3: p. 8 l. 19-20: High abiotic CO2-release may point to carbonates being present in the soil solution and to high pH. The authors discuss that issue on p. 10 l. 20 ff.

Remarks #2 and #3 let me recommend to provide some information on soil texture class, pH, organic C content and sampling depth for each site in the M&M section.

According to this suggestion we have included information about soil texture, organic carbon content, pH, sampling depth and also water holding capacity of the soil. The results underline the reviewers suggestions that especially the fine particle size is responsible for the higher water holding capacity of the soil in the Zygogonium crust:

P.4, L. 14 - 16: "It is situated 320 to 340 m a.s.l. and soils are acidic (pH = 5.28), mostly due to their origin from red sandstone of the early Triassic (Landesamt für Geologie und Bergbau, Rhineland–Palatinate), with a loamy soil texture, very low organic carbon content (<1%) and a water holding capacity of 40%." P.4, L. 23 - 25: "The bedrock is composed of base rich (pH = 6.81) limestone that originates from early Triassic with a coarse gravel overlay, with a loamy sand soil texture, 6% organic carbon and a water holding capacity of 30%." P.4, L. 27 – 28: "Sampling depth at both sides was between 0.8 to 1 cm."

We would like to thank the reviewer for the suggestion to provide information about the maximum water holding capacities, as this helps to understand and interpret the gained data and makes the values available for comparison with literature. According to the suggestion the maximum water holding capacities of both BSCs were added, to make the optimum water content comparable to maximum saturation situations. The values were calculated as follows: maximum water holding capacity of BSCsoil added to maximum water holding capacity of BSCorg.

P. 8, L. 13-14: The maximum water holding capacity was 3.29 $\pm$ 0.89 mm for C-BSCall and 4.66 $\pm$ 1.38 mm for G-BSCall.

Remark #4: p. 10 l. 1-2: The authors state a higher water holding capacity (WHC) of the Nostoc crust than the Zygogonium crust and attribute this to exopolysaccharides (EPS), which is in full agreement with the literature. However, apart from its lower NP performance, the Zygogonium crust had higher amounts of chlorophyll (Table 1), which traditionally is interpreted as a biomass equivalent. Is it possible that high Zygogonium biomass compensates for high WHC of the EPS of Nostoc? I think that the statement of

higher WHC of the Nostoc crust could be substantiated by some experimental data, or, for example, from presenting some close-up photographs of the crusts to get a visual impression of crust development.

The referee is correct in saying that it would be only logical for Z. ericetorum to hold more water than N. commune, as the green algal crust has a higher chlorophyll concentration and therefore more biomass per area than the cyanobacterial crust. However, it is difficult to compare chlorophyll content between cyanobacteria and green algae. As we explain in our discussion (P. 11 L. 25-29), chlorophyll doesn't seem to be a suitable reference value to compare NP rates between green algae and cyanobacteria, because the current calculations exclude the phycobilisomes of cyanobacteria. Therefore, the photosynthetic active pigments in cyanobacteria are underestimated and their biomass as well, as this value is traditionally interpreted as biomass.

Additionally, the factor EPS masks the effect of more biomass generally being able to hold more water: Nostoc commune does possess very thick EPS layers, that are able to hold up to 20 – 30 times their dry weight, while Z. ericetorum can't take up as much water (SATOH, Kazuhiko, et al. Recovery of photosynthetic systems during rewetting is quite rapid in a terrestrial cyanobacterium, Nostoc commune. Plant and cell physiology, 2002, 43. Jg., Nr. 2, S. 170-176. SHAW, Eric, et al. Unusual water flux in the extracellular polysaccharide of the cyanobacterium Nostoc commune. Applied and environmental microbiology, 2003, 69. Jg., Nr. 9, S. 5679-5684.). In percent the cyanobacterial crust from Fig. 4 could hold up to 4562% H2O compared to its dry weight while having 88% of its maximum NP rate. The green algal crust could only hold 435% H2O compared to its dry weight while having 18% of its maximum NP rate. This is also now stated in the manuscript on page 8, lines 13-15.

---

## Author Comment (AC2) · 12 Dec 2017

Early successional stages of biocrusts are not only important due to the fact of being pioneer colonizers of barren habitats, but also because climatic predictions point to them as possible dominant organisms in areas under strong hydric stress, where later successional stages would not succeed. Up to the authors to use this concept in the introduction in order to remark even more the importance of their study.

We agree with the reviewer that climate change might increase the difficulty of later

developmental stages of BSC to establish in habitats with less available water and increased temperatures. It might therefore be possible that a climax community dominated by cyanobacteria and green algae will be established. Nonetheless, it still needs to be mentioned, that there are in fact studies of hot and dry habitats, where e.g. lichens or bryophytes are present as the climax community (e.g. Zedda, L., Gröngröft, A., Schultz, M., Petersen, A., Mills, A., & Rambold, G. (2011). Distribution patterns of soil lichens across the principal biomes of southern Africa. Journal of Arid Environments, 75(2), 215-220. Or Weber, B., Graf, T., & Bass, M. (2012). Ecophysiological analysis of moss-dominated biological soil crusts and their separate components from the Succulent Karoo, South Africa. Planta, 236(1), 129-139.) Nonetheless, we found the comment really helpful and are glad that this point has been brought to our attention. We will include it in our introduction as it strengthens the need for more studies similar to ours. However, it is important to mention that the BSCs investigated in this study are from temperate regions in which climate change will most probably not reduce rainfall as much as that later successional stages of BSC will disappear completely, although a change in community composition is expected. We will include a remark that explains that studies taken in habitats that are more sensitive to changes in rainfall should be investigated. It is discussed now on page 2, lines 19-24.

INTRODUCTION Page 4 lines 4-5: Please clarify this sentence. I think that authors want to say here that depending of the treatment made to the sample (sample with soil, without it, or bare soil) a different response will be found in the gas exchange experiments. But I do not understand the sentence: "We expect that the position and arrangement of the sample inside the measurement system, here a cuvette, will influence the photosynthetic values". Are the authors analyzing, at some point, how the position of the sample inside the cuvette is influencing gas exchange measurements? I think that the sentence is confusing and is not a good choice to close a, on the other hand, well developed introduction

The referee is correct in pointing out, that we did in fact, not alter the position of the

sample inside the cuvette. We have removed this sentence from the introduction and discuss the topic in greater detail in the discussion (Page 11, line 15- Page 12, line: 4).

MATERIAL AND METHODS P5, L13: Could authors provide some info about why was this set of temperatures chosen for the experiment?

The chosen temperatures are in accordance with other eco-physiological studies on BSCs (e.g. Weber, B., Graf, T., & Bass, M. (2012). Ecophysiological analysis of moss-dominated biological soil crusts and their separate components from the Succulent Karoo, South Africa. Planta, 236(1), 129-139. Or Lange, O. L., Belnap, J., Reichenberger, H., & Meyer, A. (1997). Photosynthesis of green algal soil crust lichens from arid lands in southern Utah, USA: role of water content on light and temperature responses of CO 2 exchange. Flora, 192(1), 1-15. Or Lange, O. L. (1980). Moisture content and CO 2 exchange of lichens. Oecologia, 45(1), 82-87.) Additionally, they represent the average temperature range of temperate Europe (see e.g. site Homburg in Raggio, J., Green, T. A., Sancho, L. G., Pintado, A., Colesie, C., Weber, B., & Büdel, B. (2017). Metabolic activity duration can be effectively predicted from macroclimatic data for biological soil crust habitats across Europe. Geoderma, 306, 10-17.) which the study organisms face most often during a year. We wanted to investigate a broad but realistic range of temperatures to create a very detailed response of the organisms to different climatic conditions. We have added the appropriate information on page 6, lines 3 – 5.

P6, L9: I do not see clear how a one-way ANOVA can be, at the same time, a multifactorial ANOVA. To my understanding, the authors are using a one way ANOVA with type of crust being the factor (meaning that only on efactor is being analyzed), and each of the dependent variables analyzed at each moment (NP, DR, WC…..being the variable. Is this correct? Probably just a matter of terminology but I see it a bit confusing as written now

We have corrected this sentence. We used a multifactorial ANOVA where we used NP,

DR, optimum water content range and WCP as the response variables. The explaining variables were temperature and organization form (BSCall, BSCorg and BSCsoil), or an interaction of the two variables. For light compensation and light saturation only a grouped t-test has been performed. This is now clarified on page 6, lines 16 – 19.

P6, L15: A space is needed in "bystatistically". Besides, which methodology was used to compare these limits?

Space was added. Also, the statistical method was added and described in detail (page 6, line 24-27). Here a multifactorial ANOVA was used, where the explaining variables were organization form and crust dominating species (green algae or cyanobacteria) and the response variable was optimal water content.

RESULTS P7 L23-25. After having a look to Fig. 4 I agree with what is written here, but I think that is falling in contradiction with what is written in the abstract about the issue: "and low or no depression in carbon uptake at water suprasaturation" (abstract L18). I think that the text in the abstract regarding this issue should be changed to fit more accurately what is written in results

The sentence in the abstract was rephrased. Also we rephrased "low" to "minor depression" to emphasize that we mean a depression where the NP is only inhibited slightly. Page 1, line 17-18: "Nevertheless, independent of species composition, both crust types had convergent features like high light acclimatization and minor and very late occurring depression in carbon uptake at water suprasaturation."

P7 L27-28: I think that what authors want to underline here is that C-BSC and G-BSC water content values are close between them both situations, "all" samples and "dom" samples. But as it is written now it seems that, for example, for C-BSC "all" and "dom" values are similar between them, which does not seem to be correct. Just a small correction would solve the possible confusion. This sentenced was rephrased.

P: 8, L. 11-12: "The values for optimum water content between both BSCall are close,

as well as between both BSCdom which show similar values, independent of species."

FIGURES Fig. 2. I think that both sub-graphs should be scaled equally at the Y axis in order to compare gas exchange rates between C and G crust types easily The Y axis has been changed according to the suggested amendments.

Fig. 4. Please indicate in the figure legend the amount of light used for the experiment Added the PPFD under which the water curve was measured according to suggestion of both referees.

Fig. 6. This figure is hard to follow for me. I think that the variable "effect size" is a ratio between C and G crust types calculated for "dom" and "all" samples and based on area of each sample and chlorophyll content, but I do not understand why such ratio is called "effect size". Could authors please provide more explanations about this graph? I do not understand either that bump of the effect size at 25 C for chlorophyll based net photosynthesis. I have read in different parts of the text that authors consider that net photosynthesis has not a statically significant drift with temperature on an area basis, at least for the green algae crust. Does this graph mean that temperature has a significant effect over photosynthesis on a chlorophyll basis but not on area basis? Besides, the figure is supposed to show differences in the effect size for both N. commune crusts and Z. ericetorum, but I do not see clearly which is which in the graph.

Paragraph in results (P. 8, L 25- P.9 L 8) has been rephrased. We would like to provide some more information here and explain the name "effect size" on the chart. We saw that Z. ericetorum crusts always had much lower NP rates then N. commune crusts and separated organism, even though we would expect the exact opposite, as the green algal crust had up to 181 times higher chlorophyll rates per area, which would enable the crust to assimilate much more $CO_2$ than the cyanobacterial crust. We were therefore interested if this higher NP rates of N. commune were caused by the eco-physiological features, like a CCM or caused by methodological mistakes. Therefore, we compared the NP rates of both organisms for both reference values, chlorophyll

and area. If we could detect the same pattern of much higher NP rates in N. commune in both reference values, we would prove that eco-physiology was the driving factor. What we did was: We calculated NP for N. commune and for Z. ericetorum based on area and on chlorophyll each. Then divided the NP/Chlorophyll rates of N. commune with the NP/chlorophyll rates of Z. ericetorum. Next, we did the same division for the NP rates based on area. If the resulting ratios would have the same size, reference values would have no effect on the higher NP rates, therefore only eco-physiological differences would explain the much higher NP rates in the cyanobacterial crust. Our results showed the opposite: there was an obvious effect on reference values, showing in the always higher bars in chlorophyll based NP rates. We suppose therefore, that chlorophyll calculations as they are used at the moment, are not suitable to calculate NP rates in cyanobacteria, as they exclude phycobilisomes that also are responsible for NP rates. This will result in an overestimation of NP rates in cyanobacteria: In BSC 25 °C we can see that NP based on area supports that N. commune does have a seven times higher NP rate than Z. ericetorum. But NP based on chlorophyll describes a difference of 42 times higher NP rates for the cyanobacterial crust, resulting in an overestimation of NP rates in cyanobacteria up to six times compared to NP rates based on area. We did not only do this calculation for the separated organism but for the intact BSC systems, too. The much higher value of 42 times higher NP rates if they are based on chlorophyll instead of area is also owed to a temperature dependency of Z. ericetorum that we could not detect in N. commune, increasing the effect at 25 °C. The graph does not show that there is a temperature dependency visible if the NP values are based on chlorophyll but not on area basis. It only shows that the effect of temperature is stronger on chlorophyll than on area basis, but an effect is visible for both reference values. The shape of the response is only different, because the reference values are of different suitability. The name effect size is originating from the effect that the reference value has on the NP rate, but as this will confuse readers we changed it to "Ratio of NP of C-BSC/ G-BSC".

DISCUSSION P9 L13-22: Authors discuss in this paragraph about the differences in

depression of net photosynthesis at high water content between C "dom" crusts and G "domÂ'l crusts, explaining ecologically why makes sense the fact of not finding this depression in Nostoc (C) and finding it in Zygogonium (G). After having a look at figure 4, it seems to me that there are more measured points at high water content (over 80% of maximum water content) in Zygogonium than in Nostoc (I mean, for CBSC dom it seems that there is a gap between 80% and 100% of water content). Any explanation for this? Could this affect the ecological interpretation of the depression of net photosynthesis at high water contents or authors are using other indicators to analyze this issue?

The displayed data represent normalized water contents, as the absolute water content was different between both crust types and comparison therefore difficult. In the completely oversaturated N. commune crust, the amount of water brought into the measuring system was too much for the system to produce reliable measurement values. This is because of the high cross-sensitivity of the sensor between water and $CO_2$ molecules. Very high water contents result in unreliable data, as water molecules might mistakenly be detected as $CO_2$ molecules. Due to the mentioned system limitations it was impossible to measure higher $H_2O$ contents in N. commune. This is clarified on page 8, lines 6 – 8.

P9 L23-27: I have gone to the supplement figure S2 in order to try to follow the detection of the CCM mechanism and its relationship with depression of photosynthesis at high water contents. This is something quite interesting physiologically under my pointy of view that deserves more research efforts in the literature. I have seen that authors propose (correct me if I am wrong) that the fast changes in differential $CO_2$ response in the gas analyzer after light changes supports the existence of the CCM in Nostoc, and that this was not found in Zygogonium. Do you mean that the response of Zygogonium after light changes was different or somehow slower that in Nostoc? Is there any support in the literature for this pattern? (I mean presence or absence of CCMs in cyanobacterias Vs green algae)

We are glad to provide some in depth explanation on the carbon concentrating mechanism and how it was detected in N. commune but not in Z. ericetorum. In general, it is known from literature that most green algae as well as all cyanobacteria do possess an inorganic CCM (Raven, J.A., Cockell, C.S., De La Rocha, C. I. The evolution of inorganic carbon concentration mechanisms in photosynthesis. In: Phil. Trans. Soc. B. (2008)). Although the mechanisms have multiple evolutionary origins, the function is the same: CCMs accumulate $CO_2$ around rubisco. While the mechanisms behind the accumulation might be different, the photosynthetic response is the same, which can be seen in supplement figure S2 (a): There is a strong peak in carbon uptake as soon as the light is turned on, which flattens itself after a few minutes into a straight line. Usually the uptake of $CO_2$ during photosynthesis looks like a sudden drop of the $CO_2$ concentration in the measurement system gas. Afterwards the assimilation curve stays on the same level. This can be seen in S2 (b), in the downward curve just before the black arrow marks the peak in the upwards curve. If a CCM is present, this pattern is changed. As soon as the light is turned on more $CO_2$ is accumulated than would normally be the case under continuous conditions of water content, light and temperature. This is because the reservoir around rubisco is filled up, which can be seen as a sudden peak in the picture S2 (a; marked by a black arrow). As soon as the light is turned off again, $CO_2$ that has not been used during photosynthesis is released again, which is shown with a sudden increase of $CO_2$ in the measurement system gas. Here the same applies: More gas is released than normally would. After a few minutes this peak drops again, under light and in dark conditions and a continuous respiration or assimilation can be detected. We were unable to detect the same pattern in the green algae BSC, even under heavy manipulation of the measurement conditions, which included different temperatures, water contents, PPFDs and time intervals of measurement. Therefore, we conclude that no CCM can be detected in Z. ericetorum. As this was the first study to test this for this species, we provide a first insight in how this green alga photosynthesizes. This has been clarified in the caption of Table S2.

P10 L1-2: I have been following with interest the lack of optimum temperature for net photosynthesis in the green algal crusts because it was something initially unexpected to me. What I see in relation to this in Fig. 3 regarding C and G "dom" subgraphs, is that Nostoc follows a pattern of raised net photosynthesis with temperature through all the temperature range and that Zygogonium shows a raise up to 17 C and a decrease at 22 C (but 22 showed highest photosynthesis compared with 12 C). I know that authors are supporting their idea of lack of temperature optima in the stats, which I think that is right and interesting, but after looking the graphs it seems to me that it could be perfectly said that Nostoc dom has a temperature optima at 25 C and Zygogonium at 20 C. If Zygogonium is less adapted to long activity periods than Nostoc, I would expect a concentration of metabolic activity during softer environmental conditions, and this should shift temperature optimals to lower values rather that erase the concept of optimum temperature for net photosynthesis. On the other hand, author0s statement of lack of temperature optima in the green algae is supported with the graph 3b for GBSC all, where the link between net photosynthesis and temperature is clearly erratic and defined by a lack of pattern. I just would like to know author0s opinion about this, because their approach to T optima concept based in stats is absolutely right to me.

The reviewer has mentioned an interesting point here. It is absolutely possible and likely that Z. ericetorum might show an optimum temperature point somewhere between 17 °C and 25 °C, although we can only assume a trend here. As this interval with 8°C is quite broad it would be very interesting to include this temperature. A lower optimum temperature for Z. ericetorum compared to N. commune would still be in accordance with our theory, that N. commune is wet and active at higher temperatures then the green alga. Additionally, it points towards the algae being able to photosynthesize at high temperatures (for Europe; see climatic data in e.g. site Homburg in Raggio, J., Green, T. A., Sancho, L. G., Pintado, A., Colesie, C., Weber, B., & Büdel, B. (2017). Metabolic activity duration can be effectively predicted from macroclimatic data for biological soil crust habitats across Europe. Geoderma, 306, 10-17.). As temperature curves of green algal dominated BSCs are quite rare, this should be done in

future studies.

And this is a different issue, but it is surprising to me the lack of statistical differences in Nostoc between C-BSCall and C-BSCsoil net photosynthesis. It means that the photosynthetic cyanobacteria layer of the soil is not creating any relevant C input compared with bare soil. Interpretations for this behaviour?

The p-value responsible for this similarity is not very far from a statistical difference (p= 0.089400). We suppose that the high natural variation that has been shown in the high standard deviations of NP is responsible for this similarity. Increasing the sampling size would most likely result in a statistical difference between net photosynthesis input in C-BSCall and C-BSCsoil. As for now, as we cannot detect differences, we have to assume that 1) the low biomass of this very young and not diverse BSC is responsible for this low NP, or 2) that only under optimum conditions a difference can be detected, or 3) that area might not be a suitable reference value to calculate NP rates, although chlorophyll would still not be the better choice here, as it would overestimate NP rates of cyanobacteria and result in a difference that is not real.

---

## Author Comment (AC3) · 12 Dec 2017

Comments to the Authors The paper of Szyja et al. aims to characterize ecophysio­logically early successional biological soil crusts in heavily human impacted areas. For achieving this they choose two locations with a different type of BSC: one dominated by a cyanobacteria and the other by a green alga. Overall, I found the paper represent­ing an interesting contribution to scientific knowledge of BSC ecophysiology because: 1- there are at present not many data available about ecophysiology performance of these type of BSCs and 2- The comparison of the response between bare soil, intact

[Figure]

BSC and isolated compo- nent is novel and very interesting. Nevertheless, I found some important problems as how the work is presented. The main problems are in the methodology where the experimental design (mainly number of replicates in each experiment) is not clear and in the results, where some of the figures are quite confusing. The question about whether the NP rates should be expressed on a chlorophyll or surface basis is not relevant here and, obviously, will differ if comparisons are made between cyanobacteria and green algae. In my opinion the number of references (85) exceeds the needs of the paper. Beside some minor/typographic errors (i.e. check subscript in $CO_2$ throughout the text), in general, the paper is well structured, the discussion is good and conclusions clear but it needs to show results in a way that they appear more conclusive. In conclusion, I find the paper interesting and scientifically sound but taking into account the amount of data and how they are presented I don't think it reach the standards of BG. I have some comments and suggestions that I think will improve the paper.

Major and minor comments

TITLE I suggest removing the second part of the title (implications for conservation and succession) as it does not reflect the content of the paper. Second part of the title has been removed. We agree with the referee that it has no connection to the contents of the paper.

ABSTRACT There is no reference in the abstract to one of the main points in the work that is the differences found between response of intact BSCs and of it isolated dominant components. We agree with the referee that this topic was not getting enough attention in the abstract. We have included information regarding this topic in the actual version and also moved this section to end of the abstract to underline its significance for the interpretation of the data (lines: 21-24, page 1).

Page 1. Line 20. I suggest to remove the sentence beginning "Nevertheless, a major. . ." See comment above. This is a response to the general remark throughout the

referee's comment about removing the part of the study where we investigated differences in NP rates if the used reference value is either the chlorophyll content or the sample surface area.

We consider this partial aspect of the study worth mentioning, even though these coherences are well known for eco-physiology experts. Nevertheless, by being a potential contribution to the special issue on biocrusts in Biogeosciences, a major benefit of this manuscript is to reach a broad biocrust readership and also non-physiological experts. We see this as an opportunity to introduce and explain this topic to a new audience, especially because the choice of reference value is variable, depending on the investigated organisms and research question. Basing NP rates on chlorophyll content will result in an overestimation of NP rates of cyanobacteria dominated BSC compared to other crust organisms or biocrust types. We agree with the reviewer, that a comparison of gas exchange rates between different publications was, of course, not the main goal of this study. Nonetheless, we want to provide a suggestion on how to avoid discrepancies in interpretations of gas exchange data. In the actual version of the manuscript we have taken great care to clarify this point and give explanations as to how our findings may influence study design and data evaluations of similar studies in the future (Page 12, Lines: 5-20).

INTRODUCTION

Page 1. Line 29. Please rewrite the sentence "Investigations. . ." As it is now is contradictory. Are there abundant or few investigations in cyanobacteria?

Sentence has been rephrased so that it is easier to understand now (P 2-3, L 34 - 4 and P 2, L. 1-3).

Page 2. Lines 5 to 20. In my opinion the concept of arrested succession should be introduced at the beginning so it is clearer for the reader.

We have restructured and reorganized the paragraph. The concept of arrested succession is now presented in a clearer way (P 2. L 9-18:)

Page 3. Line 9. Reference Reisser et al. 2007 is not in the list.

The reference Reisser et al. 2007 has been corrected to Reisser, 2007.

Line 23. I suggest to change "or" for "and a"

Redrafted.

Lines 25-26. Were these "in situ" measurements carried out? I think it would be better say "would allow"

Was corrected.

Line 32. Colesie et al. 2014b not in the list. "Higher" than what?

Colesie et al. 2014b was corrected to Colesie et al. 2014 (without b).

Sentence has been rephrased to provide a comparison: P. 3 L. 32-33: "A higher physiological flexibility is predicted for cyanobacteria and green algae compared to bryophytes and lichens which would enable both organism groups to cope with a wider range of abiotic stresses. "

Page 4. Line 4. The sentence is confusing and I think is not relevant here. I assume that when authors refers to system they refer to BSC and not to the measurement systems. The treatment or position in the cuvette is another question. Of course there will be variability between samples, but here the comparison is between isolated individuals (green algae or cyanobacteria), soil biocrust and soil. I suggest removing this sentence.

As suggested by referee #2 and #3 this sentence has been removed and the topic is now discussed in the discussion section (P. 11 -12; L 15 – 4).

MATERIAL AND METHODS

Page 5. Line 2. Check reference Honegger 2008. Is 2003 and also it refers to green

algae photobiont but not to cyanobacteria.

Reference Honegger has been removed. Reference for N. commune (Tamaru et al., 2005) and green algae (Seckbach, 2007) have been included (P. 5, L. 3).

Line 11. n=6. It is not clear to me how the sampling or subsampling was made. From each 6 of C-BSC and 4 of G-BSC you take 3 subsamples?

We agree with the reviewer that this section in the methods was not written clearly. We have rearranged the whole section and tried to clarify terminology as well as the description of the different measurement series and units (P.5, L 32- P.6 L 3).

Line 12. First, you need to indicate how the saturation light was determined.

We rearranged the methods part and put the determination of saturation light before the determination of water dependent photosynthetic response (P.5, L. 17-31).

Line 16. Delete "from the"

Deleted.

Line 19. Should not be a new paragraph.

Deleted.

Line 21. I understand that the weighing was during the dehydration cycle to have the full response, but not between them. Please explain this.

Because of the sample being located in a closed, gastight cuvette "during" the $CO_2$ exchange measurement, it can only be weighted once this reading is taken and the cuvette is open. For detailed description please see: Photosynthesis and Water Relations of Lichen Soil Crusts: Field Measurements in the Coastal Fog Zone of the Namib Desert; O. L. Lange, A. Meyer, H. Zellner and U. Heber; Functional Ecology; Vol. 8, No. 2 (Apr., 1994), pp. 253-264. We have tried to clarify this in the completely rewritten methods section (P 5 – 6; L. 30-5).

Line 23. I suggest new paragraph. "To obtain the net response to light. . ." n=3. Are the samples BSC or species individuals? It is not clear from the text and in Fig. 2 they appear as individual species measurements. In agreement with the previous comments we have rearranged this whole section for more clarity and transparency in the used methods.

Figure description was also changed. N=3 represents BSC samples, not species individuals.

Line 25. How the optimal temperature was obtain? Are there any regressions done for this? Data is not show. Please explain.

Prior to the light dependent gas exchange experiments the operation temperature was checked by testing if the light saturation point was independent of temperature, by testing if a difference was visible between 17 °C or 25 °C, with n=3 replicates each. As no difference for the light saturation point could be detected (grouped t-test; p-value for C-BSCall: 0,095; p-value for G-BSCall: 0,597), the operation light for the water dependent gas exchange experiments was therefore set to 985 $\mu$mol photons m-2 s-1 for C-BSC and 1260 $\mu$mol photons m-2 s-1 for G-BSC, which represent the results from the 25 °C measurement. This has been clarified (P. 5, L. 24-28; P 6, L1-2).

Line 29. Should not be a new paragraph.

Deleted.

Page 6. Line 10. Include "of the two types of BSC" after "levels". The analysis as it is explained is confusing as there were different number of samples and subsamples for the different experiments. For the drying curves there were 6 C-BSC and 4 G-BSC and from each of these all, dom and soil. But in the light curves there are only 3 CBSC and 3 G-BSC without distinction of components. So, I understand that BSCall, BSCdom and BSCsoil cannot always be the explanatory variables.

The reviewer is absolutely correct in pointing out that the statistical data analysis is

written in a confusing way. We acknowledge showing us were we have described it poorly. We included the suggestion of the reviewer and rewrote the paragraph about the light curves, where we only did a grouped t-test, as correctly pointed out, we could not have organization level as a dependent variable (P. 6, L. 16).

RESULTS

The adjustments of the curves in Fig. 2 doesn't look very good, especially that of Z. ericetorum, showing an increase in the response and no saturation following the points and not the line. Please check this. Also, how where the light parameters (compensation and saturation) calculated, from individual adjusted curves or from one curve? It should be explain in material and methods. There are no supplementary tables or graphs showing values of these parameters.

According to suggestion of the referee we have included the necessary information in the material and methods section (P. 5, L. 27-29) but would like to give further information for the reviewer to follow our argumentation: The mathematical formula used to fit the curve is the so-called Smith function. It is the standard curve to fit light curves of BSC organisms as it : "makes it possible to calculate apparent maximum quantum yield of $CO_2$ fixation ($\Phi$, initial linear slope of light response curve), NPmax (the theoretical maximal rate of NP at saturating PPFD), PPFDsat (the light intensity allowing 90% of NPmax which represents a realistic estimate for light saturation [...], and PPFDcomp (the light compensation point of $CO_2$ exchange)" (from Lange, O. L., Belnap, J., & Reichenberger, H. (1998). Photosynthesis of the cyanobacterial soil‐crust lichen Collema tenax from arid lands in southern Utah, USA: Role of water content on light and temperature responses of $CO_2$ exchange. Functional Ecology, 12(2), 195-202.) The calculation for the compensation and saturation point were as follows: The original data from the GFS 3000 work sheet were put into the graphics program Sigma Plot. These data only had the following light intensities with corresponding NP rates: 0, 25, 50, 100, 300, 500, 1000, 1500, 1750, 2000 photons m-2 s-1. Calculating light saturation and compensation points from this data set is not very accurate, therefore a curve

fitting is being done with the smith function. The smith function provides 259 points for our curves interpolated between the actual measured points. Light compensation is calculated by creating a linear regression line from the last negative to the first positive point, that is created by the smith function. With the obtained formula for the slope the intersection with the x-axis can be calculated, which represents the light compensation point, where respiration equals assimilation. Light saturation is done by calculating 90% of NP and looking in the fitted curve for the NP value closest to the NP90%. The corresponding, calculated light value is then considered the light saturation. Each light curve measurement produces one light saturation and one light compensation point. The given values (P. 7 L. 4-11) were means of three measurements each. To simply say that the Smith function is the standard tool to use for BSC light curves is not sufficient, so we would like to provide the $R^2$ values of the single curves, to proof that the formula is fitting for the data: C-BSC: 0.98; 0.98; 0.98; G-BSC: 0.95; 0.97; 0.97.

Page 6. Line 4. From Figs 4a and b they don't contribute to NP response.

That is correct and exactly what we wanted to proof. We wanted to remove the soil crust organisms and only measure the organisms and soil separate. With not having many photosynthetic active cells in the soil we also did not expect a contribution to NP response.

Line 22. Here it is said G-BSCall and C-BSCall but not in Fig. 2. Please clarify. Im suggest changing "almost twice as much" for "higher"

Rephrased.

See comment above, description of light curve was clarified, description of Fig. 2 has been clarified also.

Line 24. Why organisms? Is it not BSC? It is not reasonable that the difference in compensation point was twice as much but then there were no significant. As comment above please explain how this analysis was done.

Corrected to BSC. Analysis was explained more in detail in material and methods as suggested by the reviewer. We want to emphasize that one of the main conclusions of the study is that these BSCs (and mainly the separated dominant organisms) show a very broad range of responses to different environmental parameters, also to light. In this case it means that the standard deviations are so big that even though the values for G-BSC are higher than for C-BSC, there is no statistical difference between the means.

Line 26. The same discussion will apply for the saturation points. From Fig. 2 we can understand that there is no saturation at 2000 $\mu$mol (just a few lines before it is said that maximum NP rates were reached at 2000 $\mu$mol).

Sentence was rephrased, and the following information added: The GFS3000 cannot increase the PAR above 2000 $\mu$mol photons m-2 s-1. (P. 7, L. 5). Therefore, we need to conclude that the highest NP rates we can measure are at 2000 $\mu$mol photons m-2 s-1. Additionally, the saturation point is calculated as 90% of NP (see P. 5 L. 27-28) and is not reflected by the slope of the curve.

Page 7. Line 2. Include "dominated BSC" after "commune". Refer here to Supplementary tables. Line 6. Include "dominated BSC" after "ericetorum". Refer here to Supplementary tables.

Included both. We refrained from putting the sentence in P. 7 L. 22 at the end of each paragraph and left it at the end of the segment.

Line 8. Delete "an" Water dependent photosynthetic response. In my opinion better than exemplifying graphs, average data of all replicates should be represented. Differences between just two samples are not relevant. Also, curves shown in Figures 4 are very difficult to understand as it is not normal the fluctuation around 80% water content. It must be an artifact that could be masked using averages. Also the water depression is not clear.

[Figure]

Using exemplifying graphs for presenting water dependent photosynthesis data is a common and well-accepted style in BSC, lichen and bryophyte physiology literature. Many studies on BSC, as well as lichens make use of this, because it allows a better understanding of the processes and clearer graphical design. One complication that comes with mean values plotted in these graphs, is that they vary not only along the y- but also along the x-axis, which would require a demonstration of standard deviations in two directions, which is clearly complicating the graph and the message that should be transported with it. Rather than "masking" effects, as suggested by the reviewer, by using mean values this is an option to precisely describe and demonstrate processes that are otherwise easily overseen. Plotting the response curves against a standardized water content scale (%) allows comparisons between samples with highly divergent water contents and is a tool used in review articles and book chapters. As an example to show that even overlaying single curves will not produce a clear picture, we plotted 5 NP curves of C-BSC (all, dom and soil; 25°C) on normalized water content (Fig.1). In this graph we used normalized NP rates instead of total NP rates, as we have strong variations between samples. It is obvious that all the curves for C-BSCdom are very similar, but the ones in C-BSCall vary so much that a clear pattern cannot be shown with this kind of graph. This is because of the strong variations in the soil (shown in C-BSCsoil).

In order to clarify which water contents were optimal (90% NPmax) we have included highlights in the graph and provided a table with the mean water contents after which NP is slightly inhibited (below 75%). Selected literature regarding this topic and showing the same type of graphs: - Lange, O. L., Belnap, J., & Reichenberger, H. (1998). Photosynthesis of the cyanobacterial soil‐crust lichen Collema tenax from arid lands in southern Utah, USA: Role of water content on light and temperature responses of CO2 exchange. Functional Ecology, 12(2), 195-202. - Lange, O. L., Büdel, B., Heber, U., Meyer, A., Zellner, H., & Green, T. G. A. (1993). Temperate rainforest lichens in New Zealand: high thallus water content can severely limit photosynthetic CO 2 exchange. Oecologia, 95(3), 303-313. - Lange, O. L., Green, T. A., & Heber, U. (2001). Hydranone

tion‐dependent photosynthetic production of lichens: what do laboratory studies tell us about field performance?. Journal of Experimental Botany, 52(363), 2033-2042. - Green, T. G. A., Nash III, T. H., & Lange, O. L. (2008). Physiological ecology of carbon dioxide exchange. Lichen biology. Cambridge University Press, Cambridge, 152-181. - Lange, O. L. (2001). Photosynthesis of soil-crust biota as dependent on environmental factors. Biological soil crusts: structure, function, and management, 217-240. - Green, T. A., & Proctor, M. C. (2016). Physiology of photosynthetic organisms within biological soil crusts: their adaptation, flexibility, and plasticity. In Biological Soil Crusts: An Organizing Principle in Drylands (pp. 347-381). Springer International Publishing.

Line 16. Change Table for Tables.

Changed.

Lines 26-28. Data shown in the text of ranges of optimal WC seem different from the ones in Fig. 5 (i.e. upper limit never coincident). Please check.

The referee is absolutely correct. We accidentally used the wrong values in the text, although discussing the correct ones and also doing the statistical analysis with the correct values. We corrected the text passage (P. 8, L. 10-15).

Page 8. Line 5. I would rather delete this subsection as discussed above.

We would like to include this topic, as explained in the comment above.

Line 20. Table S6

Changed.

DISCUSSION

Line 25. BSCs photosynthetic organisms

Changed.

Page 9. Line 5. Delete "none" and better G-BSCall and C-BSCall. What does it means

physiological flexibility to water gain?

We have clarified the interpretation of this result and explain now that: "both organism groups take the same functional role in the BSC consortium and can operate at near optimal conditions over a variety of different water contents, as it would also be expected for highly stress tolerant crust pioneer species" (P. 9, L. 26-30).

Line 18. I suggest delete sentence beginning "A depression. . ." as it has already said before.

Deleted.

Line 23. I don't see the detection of a CCM from Fig.S2.

Referee #2 also asked about an explanation as to how we were able to detect that N. commune had a CCM and Z. ericetorum did not show one. We will copy part of the answer here, as it also describes how a CCM can be seen in gas exchange data: In general, it is known from literature that most green algae as well as all cyanobacteria do possess an inorganic CCM (Raven, J.A., Cockell, C.S., De La Rocha, C. I. The evolution of inorganic carbon concentration mechanisms in photosynthesis. In: Phil. Trans. Soc. B. (2008)). Although the mechanisms have multiple evolutionary origins, the function is the same: CCMs accumulate $CO_2$ around rubisco. While the mechanisms behind the accumulation might be different, the photosynthetic response is the same, which can be seen in supplement figure S2 (a): There is a strong peak in carbon uptake as soon as the light is turned on, which flattens itself after a few minutes into a straight line. Usually the uptake of $CO_2$ during photosynthesis looks like a sudden drop of the $CO_2$ concentration in the measurement system gas. Afterwards the assimilation curve stays on the same level. This can be seen in S2 (b), in the downward curve just before the black arrow marks the peak in the upwards curve. If a CCM is present, this pattern is changed. As soon as the light is turned on more $CO_2$ is accumulated than would normally be the case under continuous conditions of water content, light and temperature. This is because the reservoir around rubisco is filled up, which can

be seen as a sudden peak in the picture S2 (a; marked by a black arrow). As soon as the light is turned off again, CO2 that has not been used during photosynthesis is released again, which is shown with a sudden increase of CO2 in the measurement system gas. Here the same applies: More gas is released than normally would. After a few minutes this peak drops again, under light and in dark conditions and a continuous respiration or assimilation can be detected. We were unable to detect the same pattern in the green algae BSC, even under heavy manipulation of the measurement conditions, which included different temperatures, water contents, PPFDs and time intervals of measurement. Therefore, we conclude that no CCM can be detected in Z. ericetorum. As this was the first study to test this for this species, we provide a first insight in how this green alga photosynthesizes.

Additionally, we want to provide the publication where the method of detection was used for the first time: Badger, M. R., Pfanz, H., Büdel, B., Heber, U., & Lange, O. L. (1993). Evidence for the functioning of photosynthetic CO2-concentrating mechanisms in lichens containing green algal and cyanobacterial photobionts. Planta, 191(1), 57-70 We have clarified this topic in the text of the supplement material in order to make the phenomenon understandable for a broad readership (Figure description of S2).

Page 11. Lines 2 and 7. Species name in italics.

Changed.

Lines 16-26. In my opinion this question is not relevant as it is obvious.

We would like to refer to our comment at the beginning of this letter, that we consider discussing the differences between these two options is an important information and of interest for a broad BSC readership.

CONCLUSIONS

Page 12. Line 3. The authors conclude that there is a relative temperature independence of NP but the results show significant differences in the response of NP to

temperature.

Here we disagree with the referee, according to the statistical tests, there is no significant difference in the response of NP to temperature, except for the one mentioned by us (25°C for C-BSCdom C-BSCall and C-BSCSoil).

Line 5. In general, the question about physiological plasticity should be avoid because there are no experiments proving this.

We agree with the reviewer that plasticity was not measured in the presented study and rephrased the sentence for clarity and a more precise ecological interpretation (P. 12, L. 24-26).

Line 6. To incorporate the results into global scale carbon cycle models, the work should better provide numerical data sets (i.e. tables).

Part of the publication process for BGS is to upload the original data to an online database, so that it can be accessed easily, therefore the numerical data of this publication will hopefully be made available.

REFERENCES There are too many for the paper. As mentioned above some cited literature in the text is not in the list. Please check references through the list.

Checked and corrected the literature to fit the text.

Some literature has been removed, as single cited publications behind some statements should be enough. Still, the number of publications did not decline a lot, also because we needed to include some sources because of added comments from the other referees.

Table 1. Following my suggestion about Chlorophyll question then this should not be included.

We would like to include this part of the publication, but we put it in the supplemental material as Table S10. See comment at the beginning.

Figure 2. Legend.The second sentence is not necessary, just n=3.

Removed.

Figure 3. Legend. What do you mean by . . .of one of the group only? Please indicate what vertical bars represent.

"Of one group only" has been corrected to "organization levels", to make it clear that one BSCall, BSCsoil or BSCdom was being compared. Also vertical bars were described as being standard deviations.

Figure 4. See comment above. Indicate PAR

This has been changed.

Figure 5. This graph is very difficult to understand. See comments above. What does the letters mean? Why n=24 here?

The letters on the graph represent statistical differences between each lower limits (letters a-c) and between all upper limits (d-g). As described in p. 6 L. 24 -27: the optimum water content was compared by statistically testing if the upper and lower limits between the both BSCs and their components differed. This means we compared if the lower limits differed from one another, then we compared if the upper limits differed. We have now included a clear definition of optimum water content (water content at which 90% of NP is reached to water content at which it decreases below 90% again). The optimum water content is calculated independent of temperature, as it is not influenced by temperature. Therefore, we have pooled all readings (6 samples times 4 temperatures) for C-BSC. The sampling size for G-BSC was missing, which is 16 (4 samples times 4 temperatures). We clarified this in the text (P. 6, L. 6):

Figure 6. As suggested above I would not include this graph.

We would like to include this part of the publication. See comment at the beginning.

[Figure]

**Fig. 1.** Fig. 1: Normalized NP rates plotted against normalized water content for 5 samples of C-BSC at 25 °C.

---

## Referee Report (RR1)

Manuscript bg 2017-369, final comments from referee 2:

I am happy with the answers provided by the authors and with the interesting scientific debate created around it, so I recommend publication of the manuscript in biogeosciences. Just a few comments to make in relation with the answers provided:

- Depression of NP at high WC (FIG. 4). I still do not see clearly why the depression of NP is similar, late and low in C and G crusts (current abstract). Regarding C crusts (at least for "dom" sample) authors claim that could not detect feasible measuring points between 100%-80% of normalized water content, probably a critical range to analyze the concept of depression of NP at high WC. At the same time, depression of NP at high WC in G crusts is not low under my point of view, since it starts at 0 or very close to 0 at high WC for the "all" and "dom" samples respectively

- CCM mechanism. Thanks for interesting explanations provided. I was wondering in a possible suggestion to see the pattern mentioned more clearly, same graphs than S2 could be included for the G crusts in order to show clearly differences in CO2 gas exchange patterns in a BSC with CCM Vs without CCM

- Reference list: The reference provided of Raggio et al. (2017) has a wrong title in the reference list. Authors wrote it correctly in the rebuttal to my comments but not in the reference list. Please change it

That's all

---

## Author Response (AR2)

We would like to thank the two reviewers again for their very helpful comments, which helped to improve our manuscript again. The revised manuscript includes all the points raised by the reviewers. While the reviewers' comments are shown in grey text, our responses are formatted as standard text. Line indications refer to the revised manuscript without marked changes.

**Response to referee's letter:**

**Referee #1**

Dear authors my congratulations for the improvement of the paper after the corrections/ rewritten. In the present from I would recommend publication in Biogeosciences.

Just a few corrections must be considered:

P.7-L.28. Tables S1-S3

P.17-L1. Shanhum et al. after Seckbach (please check this reference)

Both has been edited and checked.

**Referee #2**

I am happy with the answers provided by the authors and with the interesting scientific debate created around it, so I recommend publication of the manuscript in Biogeosciences. Just a few comments to make in relation with the answers provided:

- Depression of NP at high WC (FIG. 4). I still do not see clearly why the depression of NP is similar, late and low in C and G crusts (current abstract). Regarding C crusts (at least for "dom" sample) authors claim that could not detect feasible measuring points between 100%-80% of normalized water content, probably a critical range to analyze the concept of depression of NP at high WC. At the same time, depression of NP at high WC in G crusts is not low under my point of view, since it starts at 0 or very close to 0 at high WC for the "all" and "dom" samples respectively

The referee is correct in pointing out, that 80-100% of water content is the most interesting part of a water curve, related to depression because of suprasaturation. As can be seen in Fig. 4 not all samples introduced so much water to the measurement-system to make it impossible to measure high water contents. High values were measured for e.g. all C-$BSC_{all}$ samples; C-$BSC_{dom}$ samples varied – some very small samples could be measured up to higher values as they didn't introduce so much water. It was possible to calculate from all measured curves the amount of water at which point NP was going below 75% of $NP_{max}$ (Table 1). We defined this as a depression, but not a pronounced one, as only 25% of $NP_{max}$ are reduced (i.e. "minor"): (P.6 – L. 6-7): […] whereas a minor depression because of suprasaturation was defined as NP being constantly below 75% of maximum NP of the sample."

The results are considered similar, as the statistical analysis of Table 1 did not show big differences between any of the measured values (except for C-BSCall and G-BSCdom, where G-BSCdom is having the earlier depression).

Lastly, "late occurring" resulted also from Table 1: it shows that the values for a minor depression are always high (i.e. "late occurring") – for example a water content of 84% in C-BSCall and 71% in G-BSCall is needed to reduce NPmax up to 25%. We emphasized this now on P.8-L.7-8.

Even though the graphs in Fig. 4 (b; *Z. ericetorum*) all end at or below 0 µmol*m$^{-2}$s$^{-1}$ at high water contents, their reduction is only occurring at the highest values. This is in accordance with our abstract, where we point out, that the depression is in fact occurring, but it is only occurring late, is minor and similar in all investigated BSCs and organisms.

- CCM mechanism. Thanks for interesting explanations provided. I was wondering in a possible suggestion to see the pattern mentioned more clearly, same graphs than S2 could be included for the G crusts in order to show clearly differences in CO2 gas exchange patterns in a BSC with CCM Vs without CCM

According to the suggestion we have included an exemplary graph of *Z. ericeoturm*. As mentioned by the referee, this makes it much easier for an unfamiliar Reader to understand the difference.

- Reference list: The reference provided of Raggio et al. (2017) has a wrong title in the reference list. Authors wrote it correctly in the rebuttal to my comments but not in the reference list. Please change it.

Has been corrected.

[revised manuscript text omitted]

10 **Table S4: Statistical analysis of upper limits of optimum water content of both BSC-Systems and their respective separate organisms. Shown are p-values of the Ttukey post-hoc test.**

| | C-BSC$_{all}$ | G-BSC$_{all}$ | C-BSC$_{soil}$ | G-BSC$_{soil}$ | C-BSC$_{dom}$ | G-BSC$_{dom}$ |
|---|---|---|---|---|---|---|
| **C-BSC$_{all}$** | | 0.82 | 0.65 | 0.14 | 0.00 | 0.02 |
| **G-BSC$_{all}$** | 0.82 | | 0.11 | 0.86 | 0.00 | 0.00 |
| **C-BSC$_{soil}$** | 0.65 | 0.11 | | 0.00 | 0.15 | 0.38 |
| **G-BSC$_{soil}$** | 0.14 | 0.86 | 0.00 | | 0.00 | 0.00 |
| **C-BSC$_{dom}$** | 0.00 | 0.00 | 0.15 | 0.00 | | 1.00 |
| **G-BSC$_{dom}$** | 0.02 | 0.00 | 0.38 | 0.00 | 1.00 | |

**Table S.5: Statistical analysis of lower limits of optimum water content of both BSC-Systems and their respective separate organisms. Shown are p-values of the Ttukey post-hoc test.**

| | C-BSC$_{all}$ | G-BSC$_{all}$ | C-BSC$_{soil}$ | G-BSC$_{soil}$ | C-BSC$_{dom}$ | G-BSC$_{dom}$ |
|---|---|---|---|---|---|---|
| **C-BSC$_{all}$** | | 0.94 | 0.52 | 0.39 | 0.04 | 0.06 |
| **G-BSC$_{all}$** | 0.94 | | 0.14 | 0.95 | 0.01 | 0.01 |
| **C-BSC$_{soil}$** | 0.52 | 0.14 | | 0.01 | 0.81 | 0.79 |
| **G-BSC$_{soil}$** | 0.39 | 0.95 | 0.01 | | 0.00 | 0.00 |
| **C-BSC$_{dom}$** | 0.04 | 0.01 | 0.81 | 0.00 | | 1.00 |
| **G-BSC$_{dom}$** | 0.06 | 0.01 | 0.79 | 0.00 | 1.00 | |

**Table S6: Mean values of maximum respiration rate per area of soil of both study sites. Sample size is n=3 in both cases.**

|  | Parking Lot | Standard deviation | Mehlinger Heide | Standard deviation |
|---|---|---|---|---|
| Max. respiration before autoclaving [µmol/ m²s] | -0.23 | 0.20 | -0.43 | 0.24 |
| Max. respiration after autoclaving [µmol/ m²s] | -0.06 | 0.01 | -0.09 | 0.01 |

**Table S7: Statistical significance of organization level, temperature and interaction of those effects on DP of C-/G-BSC$_{all}$, C-/G-BSC$_{dom}$, and C-/G-BSC$_{soil}$**

| Effect | df | F | p |
|---|---|---|---|
| *N. commune* | | | |
| Organization level | 2 | 0.14 | 0.872 |
| Temperature | 3 | 6.27 | 0.001 |
| Organization level * Temperature | 6 | 0.12 | 0.993 |
| *Z. ericetorum* | | | |
| Organization level | 2 | 1.01 | 0.376 |
| Temperature | 3 | 2.92 | 0.047 |
| Organization level * Temperature | 6 | 0.63 | 0.705 |

**Table S8: P-values of a Ttukey post-hoc test for DP depending on temperature in a C-BSC.**

| Temperature | 7 °C | 12 °C | 17 °C | 25 °C |
|---|---|---|---|---|
| 7 °C |  | 0.994 | 0.815 | 0.001 |
| 12 °C | 0.994 |  | 0.926 | 0.002 |
| 17 °C | 0.815 | 0.926 |  | 0.011 |
| 25 °C | 0.001 | 0.002 | 0.011 |  |

**Table S9: P-values of a Tukey post-hoc test for NP depending on temperature in a G-BSC.**

| Temperature | 7 °C | 12 °C | 17 °C | 25 °C |
|---|---|---|---|---|
| **7 °C** | | 0.408 | 0.079 | 0.041 |
| **12 °C** | 0.408 | | 0.803 | 0.638 |
| **17 °C** | 0.079 | 0.803 | | 0.992 |
| **25 °C** | 0.041 | 0.638 | 0.992 | |

**Table S10: Chlorophyll content per area in *N. commune* and *Z. ericetorum* samples without soil.**

| Sample | Chlorophyll content per insolated area [mg/cm²] |
|---|---|
| *N. commune* S1 | 0.23 |
| *N. commune* S2 | 0.10 |
| *N. commune* S3 | 0.49 |
| *N. commune* S4 | 1.48 |
| *N. commune* S5 | 0.17 |
| *N. commune* S6 | 0.36 |
| | |
| *Z. ericetorum* S1 | 18.13 |
| *Z. ericetorum* S2 | 1.11 |
| *Z. ericetorum* S3 | 15.79 |
| *Z. ericetorum* S4 | 2.68 |

[Figure]

**Figure S1: Temperature dependent dark respiration per area in (a)** *N. commune* **and b)** *Z. ericetorum* **(dominated BSCs, as well as separated organism and soil. Capital letters describe significant differences in organization level between BSC, organism and soil, whereas lower case letters compare temperature differences in one of the groups only. Sample size: n=6 for** *N. commune,* **n=4 for** *Z. ericetorum***.**

[Figure]

[Figure]

**Figure S2: CO$_2$ exchange pattern of *N. commune* and *Z. ericetorum* (c) at 7 °C with optimal water content. The blue line represents relative humidity at the moment of measurement in percent, while the red line represents the difference of CO$_2$ between reference and sampling gas in the GFS 3000 in ppm. Abscise is the time in minutes. Grey underlay represents light being turned off. The black arrow marks one example of a sudden increase of CO$_2$ uptake as soon as the light was turned on (a), or release as soon as the light was shut off (b). The dotted line indicates the normal gas exchange pattern without a CCM being active in a light-dark-cycle. Here, after a sudden drop (light being turned on) or increase (light being turned off), the red line should flatten immediatley and result in a straight line, as can be seen in the pattern of *Z. ericetorum* (c). When a CCM is present, this is not the case: the solid line in (a) and (b) represents a CCM being active. Here, the uptake of CO$_2$ is much higher than the normal NP answer would be, while the line flattens itself only after a couple of minutes.**

---

## Author Response (AR3)

Dear Emilio Rodriguez-Caballero,

The units on the y-axis have been corrected in Fig. 2, 3, 4 and supplement FigS1. Also, the acronyms in tables and figures have been described.

[revised manuscript text omitted]
). The dotted line indicates the normal gas exchange pattern without a CCM being active in a light-dark-cycle. Here, after a sudden drop (light being turned on) or increase (light being turned off), the red line should flatten immediatley and result in a straight line, as can be seen in the pattern of *Z. ericetorum* (c). When a CCM is present, this is not the case: the solid line in (a) and (b) represents a CCM being active. Here, the uptake of $CO_2$ is much higher than the normal NP answer would be, while the line flattens itself only after a couple of minutes.